# Effects of public-health measures for zeroing out different SARS-CoV-2 variants

Yong Ge [1,2,3,14] ✉, Xilin Wu [1,3,14], Wenbin Zhang [1,3,4,14], Xiaoli Wang[5,14], Die Zhang[1,2], Jianghao Wang [1,3], Haiyan Liu [6], Zhoupeng Ren [1], Nick W. Ruktanonchai[7], Corrine W. Ruktanonchai [7], Eimear Cleary[4], Yongcheng Yao[4,8], Amy Wesolowski[9], Derek A. T. Cummings[10], Zhongjie Li[11], Andrew J. Tatem [4] & Shengjie Lai [4,12,13] ✉

Targeted public health interventions for an emerging epidemic are essential for preventing pandemics. During 2020-2022, China invested significant efforts in strict zero-COVID measures to contain outbreaks of varying scales caused by different SARS-CoV-2 variants. Based on a multi-year empirical dataset containing 131 outbreaks observed in China from April 2020 to May 2022 and simulated scenarios, we ranked the relative intervention effectiveness by their reduction in instantaneous reproduction number. We found that, overall, social distancing measures (38% reduction, 95% prediction interval 31-45%), face masks (30%, 17-42%) and close contact tracing (28%, 24-31%) were most effective. Contact tracing was crucial in containing outbreaks during the initial phases, while social distancing measures became increasingly prominent as the spread persisted. In addition, infections with higher transmissibility and a shorter latent period posed more challenges for these measures. Our findings provide quantitative evidence on the effects of public-health measures for zeroing out emerging contagions in different contexts.

In the early stage of epidemics, it is critical to implement precise and effective public-health measures to control the spread and contain community-level transmission in a timely manner, with the aim of preventing outbreaks from developing into a major public-health crisis similar to COVID-19 crisis[1-4]. During the COVID-19 pandemic, governments worldwide have deployed various measures including non-pharmaceutical interventions (NPIs) and vaccinations to reduce transmission across waves in 2020–2022. After the initial outbreak in Wuhan, China's zero-COVID policy, implemented from April 2020 to early December 2022, has been among the strictest, longest approaches to tackling the pandemic anywhere in the world[5,6]. This strategy for zeroing out emerging contagions aimed to fully interrupt the transmission of SARS-CoV-2 in varying-scale COVID-19 outbreaks caused by variants with different transmissibility[7]. The interventions included localized and intense public-health measures, such as contact tracing and isolation, full or partial lockdowns, physical distancing,

[1]State Key Laboratory of Resources and Environmental Information System, Institute of Geographic Sciences and Natural Resources Research, Chinese Academy of Sciences, Beijing, China. [2]Key Laboratory of Poyang Lake Wetland and Watershed Research Ministry of Education, Jiangxi Normal University, Nanchang, China. [3]College of Resources and Environment, University of Chinese Academy of Sciences, Beijing, China. [4]WorldPop, School of Geography and Environmental Science, University of Southampton, Southampton, UK. [5]Beijing Center for Disease Prevention and Control, Beijing, China. [6]Marine Data Center, Southern Marine Science and Engineering Guangdong Laboratory (Zhuhai), Zhuhai, China. [7]Population Health Sciences, Virginia Tech, Blacksburg, VA, USA. [8]School of Mathematics and Statistics, Zhengzhou Normal University, Zhengzhou, China. [9]Department of Epidemiology, Johns Hopkins Bloomberg School of Public Health, Baltimore, MD, USA. [10]Department of Biology and Emerging Pathogens Institute, University of Florida, Gainesville, FL, USA. [11]School of Population Medicine and Public Health, Chinese Academy of Medical Sciences & Peking Union Medical College, Beijing, China. [12]Institute for Life Sciences, University of Southampton, Southampton, UK. [13]Shanghai Institute of Infectious Disease and Biosecurity, Fudan University, Shanghai, China. [14]These authors contributed equally: Yong Ge, Xilin Wu, Wenbin Zhang, Xiaoli Wang. ✉e-mail: gey@lreis.ac.cn; Shengjie.Lai@soton.ac.uk

mass testing, and strict mask mandates. Given widespread vaccination efforts, the decline in clinical severity of variant strains, the high socioeconomic costs, and potential secondary health hazards including mental health[5,8–10], the zero-COVID policy has been rapidly lifted across the country since December 2022, with a surge of infections caused by the Omicron sub-lineages, BA.5.2 and BF.7[11,12]. However, the effect of measures under this policy against variants of varying transmissibility remains to be quantitatively investigated.

A considerable number of research[13–20] has demonstrated the effectiveness of NPIs in reducing COVID-19 transmission, however, previous studies have rarely focused on the elimination effects of NPIs against different variants, particularly for highly contagious pathogens like the Omicron variant[21], particularly because many countries either instated limited NPIs or exhibited poor adherence during the emergence of Omicron. As emerging infectious diseases may have different transmission routes and dynamics, public-health interventions should be tailored accordingly to the different characteristics of infections during the early stage of epidemics, which is crucial for preventing escalation to pandemic level[22–24]. However, the degree to which these

measures are effective in eliminating outbreaks of emerging respiratory pathogens with varying epidemiological features, and in different settings, is yet to be determined.

In many countries, NPI adherence reduced significantly after the initial wave in 2020 and the goal of interventions across waves changed from containment to mitigation[25–29], implying potential confounding between reduced NPI effectiveness and weakened measures. China's COVID-19 responses for different strains provide us with a unique, real-world dataset, in terms of size and duration, to assess the effectiveness of zero-COVID measures against emerging respiratory infectious diseases of different transmissibility and virulence in diverse settings (see Fig. 1). Taking the responses to SARS-CoV-2 variants in China as an example, we designed a rigorous multi-year data collection program to assemble a comprehensive dataset, describing the infection profile and countermeasures for each outbreak in China from April 2020 through May 2022. Mathematical models for simulating transmission and Bayesian inference models were built to evaluate the impact of different interventions and define which measures were most useful for eliminating emerging contagions during the early

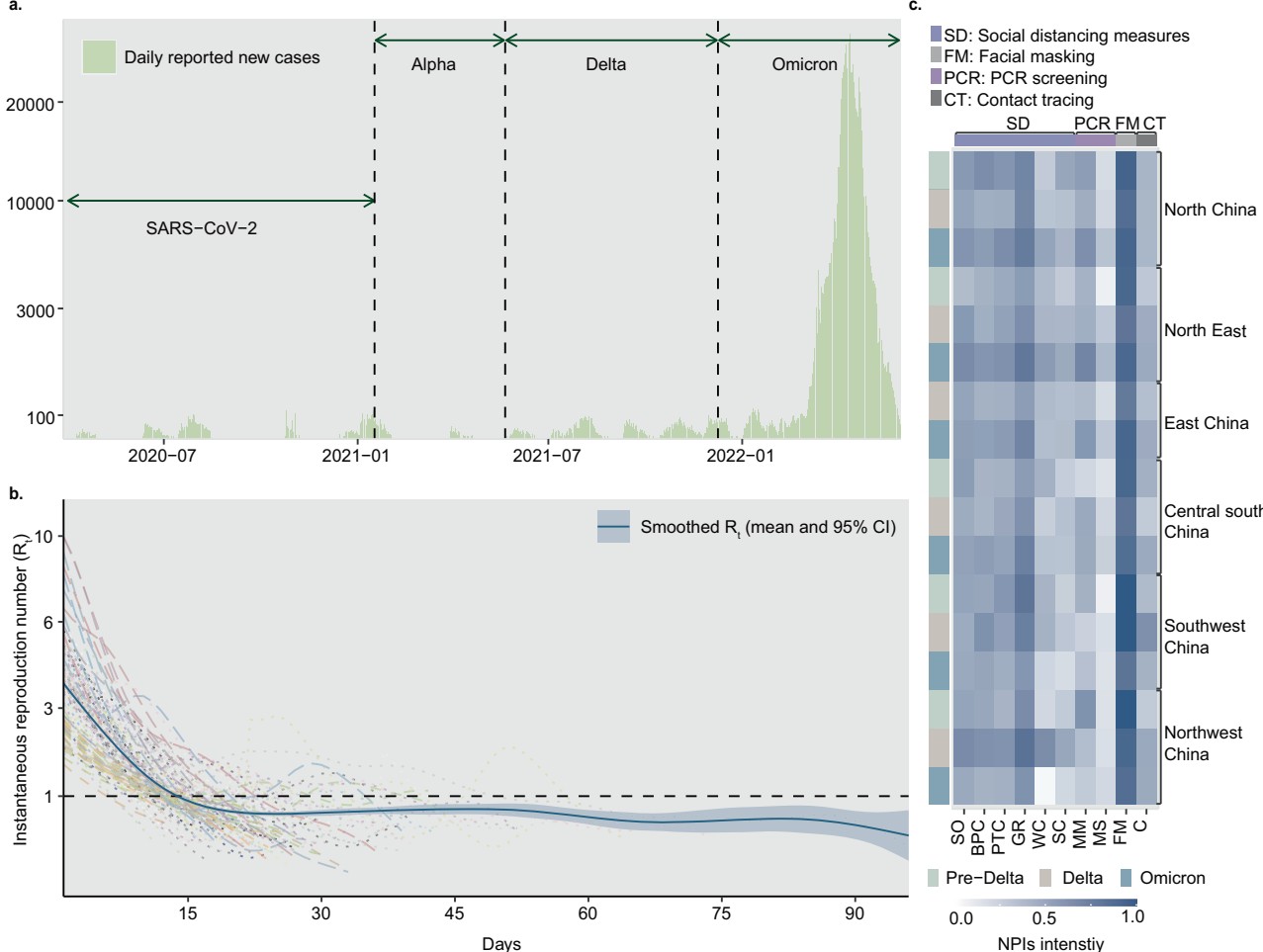

**Fig. 1 | COVID-19 outbreaks and interventions in China during zero-COVID policy under Pre-Delta, Delta, and Omicron periods. a** Daily new cases reported during the 131 outbreaks in mainland China, from April, 2020 to May, 2021. The green arrows mark the predominant strains within each stage of the pandemic. **b** Estimated instantaneous reproduction number ($R_t$) for each outbreak, aligned with the start date of each outbreak. The solid blue line illustrates the estimated overall $R_t$ for 131 outbreaks, while the blue shaded area indicated its 95% confidence interval (95%CI). **c** Heat map of mean intensity level of interventions for different variants and geographic regions of China. The color bar on the left side of the heat map represents the variant of each outbreak, green for Pre-Delta period,

brown for Delta period and blue for Omicron period. The *x* axis shows the abbreviations of non-pharmaceutical measures, including stay-at-home order (SO), business premises closure (BPC), public transportation closure (PTC), gathering restriction (GR), workplace closure (WC), school closure (SC), medicine management (MM), mass screening (MS), facial masking (FM) and contact tracing (CT). We divided the 10 NPIs into four categories: social distancing measures (SD), polymerase chain reaction screening (PCR), contact tracing (CT), and facial masking (FM). The color bar above the heat map represents the category to which each individual measure belongs.

stages of outbreaks, according to local conditions and epidemiological characteristics (see Methods).

## Results

### Effects of zero-COVID interventions against different variants in China

From April 2020 to May 2022, 90 prefecture-level Chinese cities reported 131 outbreaks that had at least 50 cases and lasted at least 7 days (Fig. 1 and Supplementary Fig. 1). These included 12 outbreaks caused by the pre-Delta variants (i.e., lineages B.1.1, B.2, and B.1.1.7), 27 outbreaks of the Delta variant, and 92 outbreaks of the Omicron lineages. The implementation of zero-COVID policy was heterogeneous across regions and over time (Fig. 1c). For example, on a regional average, the intensity of stay-at-home order in South Central and Southwest China decreased from 0.5 in Pre-Delta era to 0.4 in Omicron era, while mass screening had an intensity increasing from 0.1 to ~0.3 in most regions (Note that the intensity of each measure was normalized from 0 to 1, where 1 indicates the strictest and 0 indicates the least strict). This indicates that affected cities in southern China eased some stringent measures, such as stay-at-home order, and increased the frequency of PCR testing. Based on the data of these outbreaks and Bayesian inference models with a leave-one-out cross-validation approach[30], we estimated the effects of four groups of interventions, i.e., social distancing measures, nucleic acid PCR screening, contact tracing, and facial masking, under the policy for fully interrupting the transmission in local communities. As Fig. 2a shows, overall, social distancing measures, through reducing the spread caused by human movements and physical contact, had the most desirable effect on the reduction in $R_t$ (>38%), while facial masking (mean 30%, 95% prediction interval 17–42%) and contact tracing (28%, 24–31%) showed a lower impact, respectively.

Social distancing measures similarly showed a strong ability to reduce transmission before the Delta-variant outbreaks (over 50%). However, during the Delta and Omicron outbreaks, social distancing measures with a 30% and 33% reduction in $R_t$, respectively, were relatively less effective at preventing transmission than they were before the Delta era. For facial masking, the lowest contribution to transmission reduction (24%, −1%–60%) was shown in the pre-Delta era. In the Delta era, it demonstrated a moderate $R_t$ reduction of 43% (20%–64%), while in the more recent Omicron era, it showed the highest effectiveness of 53% (32%–64%). In addition, contact tracing demonstrated the highest ability to reduce transmission (24%, 0%–47%) after the emergence of the Omicron variant, ranking second among the original and Alpha strains (12%, 0%–46%). Mass PCR screening showed various abilities to reduce $R_t$ among pre-Delta (11, 0%–45%), Delta (3%, −1–15%), and Omicron era (2%, −1–13%).

Nevertheless, when the 131 COVID-19 outbreaks were divided into four groups based on infectivity and duration, we found that PCR screening, which focused on detecting each infected individual in communities, was more effective in combating outbreaks that had sustainable transmission. Contact tracing played a critical role in containing outbreaks in the early stages, particularly for small outbreaks (32%, 95%CI 28–35%). However, as transmission continued, the relative impact of contact tracing decreased (from 32% to 2%), while social distancing emerged as the most effective measure (from 34% to 62%). More details can be found in supplementary Section B.7.

To further validate the effects of NPIs estimated by Bayesian inference, we employed an Intervention-SEIR-Vaccination (ISEIRV) model to simulate transmission under varying real-world and counterfactual intervention scenarios (e.g., without implementing one or all NPIs, see Fig. 2b). We found the implementation of NPIs protected >98% of the would-be infected populations from infection in each city. The prompt implementation of, and stringent adherence to, NPIs protected 80 (95% confidence interval [CI], 55–110) million people from infection during the same periods of real-world outbreaks in the affected areas. Furthermore, we evaluated the contribution of each NPI in reducing infections under different variants by simulating transmission in the absence of NPI implementation. The results corroborate the ranking of NPI effects inferred by the Bayesian model (Fig. 2).

However, there are two key aspects that need to be considered when interpreting the above findings. First, this study did not assess the effects of long-lasting international travel restrictions and quarantine for reducing the introduction risk, which might overestimate the impact of other interventions in containing local transmission within each city. Second, the small-scale, short-duration outbreaks were excluded from the modeling, which might have led to an underestimation of the effectiveness of some NPIs, such as contact tracing, that might also play an important role in controlling epidemics in the early phases.

### Impacts of timing and intensity on NPI effects

To further understand how the timing and intensity of various NPIs shaped their effectiveness in containing emerging infections under different settings, we ran simulations for different cities, including five large population-size cities (LC), five medium population-size cities (MC), and five small population-size cities (SC), where Omicron outbreaks occurred (Supplementary Table 9). Each group of simulations contained transmission curves for 1800 scenarios, of which four NPIs with ten levels of intensity were implemented at five different start times under nine combinations of propagation parameters ($R_0 = 3$, 8, and 13; Latent period = 1, 4, and 7 days). Fig. 3 shows the relative reduction of daily mean infections for each scenario in different population-size cities, relative to the baseline scenario without any interventions implemented.

We found universally that the sooner all NPIs were in place, the more effective they were. The more contagious the pathogen and the longer the latent period was, the more necessary it was to introduce NPIs as early as possible. Meanwhile, a smaller population means a shorter window of timing for NPI implementation, as the susceptible population would be depleted more quickly in smaller cities. Among the four NPI groups, contact tracing proved to be the most effective measure across city sizes, showing more effectiveness at intensities above 0.6, 0.5, and 0.4 in LCs, MCs, and SCs, respectively. However, when the start time exceeded a certain threshold (~25 days), the effect of contact tracing fell dramatically even at the highest intensity across both MCs and LCs. The second-ranked intervention was social distancing measures, particularly in containing the spread of infectious diseases in LCs. We also found that a higher intensity of NPI was necessary for diseases with higher transmissibility and shorter latent periods. Facial masking showed similar effects among all cities, but the effect of large-scale testing diminished as the population increased.

### Optimal intervention packages and challenges in zeroing out emerging infections under varying scenarios

To seek the optimal combination of interventions, we simulated the infection profile with two or more NPIs implemented simultaneously under different outbreak parameters. As illustrated in Fig. 4 and supplementary Section D.1, we traversed each combined scenario of intervention strategies as well as the start time and intensity in LC, MC, and SC settings. Since the long-term implementation of NPIs usually requires enormous resource and livelihood sacrifice[31,32], a simulated policy scheme that failed to stop the transmission within 3 months (90 days) or where the outbreak infects >10,000 people was deemed ineffective, considering the policy's potential socioeconomic and secondary health impacts (e.g., overwhelmed healthcare)[33–35].

We found that implementing NPIs after day 14 of the outbreak would fail to interrupt transmission within 60 days when $R_0$ exceeded 8. When $R_0$ was <3, contact tracing coupled with a mask-wearing order can effectively contain the outbreak within 60 days, regardless of a

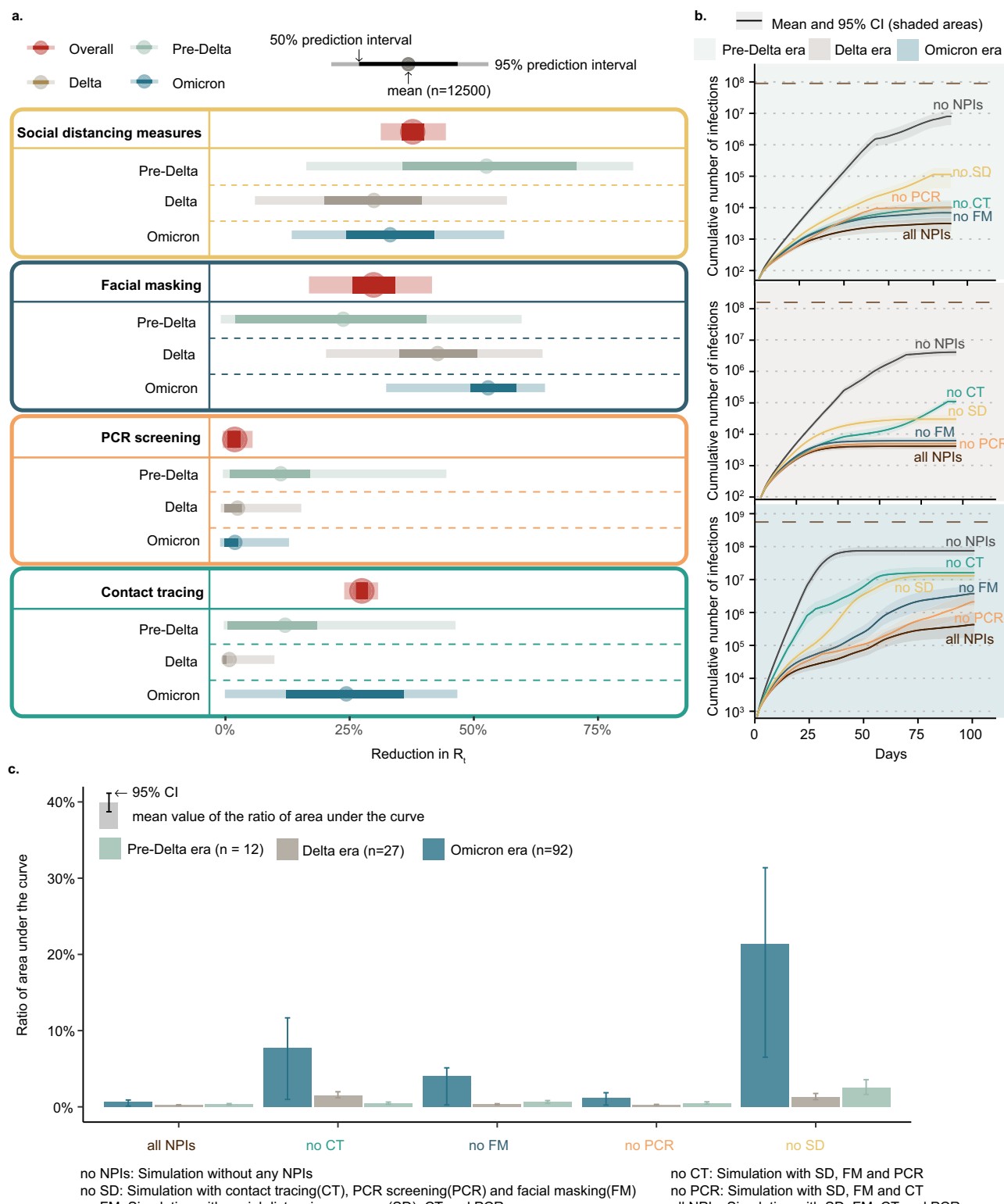

city's population size. As the latent period grew or the population size increased, the time cost of zeroing infections would increase concomitantly. For all cities, tracing and isolating at least 60% of close contacts by day 7 of outbreaks was generally effective. However, if the start of intervention was delayed until day 14, the intensity of contact tracing might need to be increased to 70% contacts traced for MCs and 80% contacts traced for LCs.

Containing infectious diseases with higher transmissibility and shorter incubation period could be particularly challenging using NPIs alone. To effectively prevent an outbreak ($R_0 = 13$ and latent period = 4) from escalating into pandemic, cities might start interventions within a week of the outbreak, coupled with efficient close contact tracing (>70%) and relatively rigorous social distancing measures (>0.5). For areas with more dense populations (e.g., LCs), the strictest

**Fig. 2 | The relative effects of interventions in containing different SARS-CoV-2 variants. a** The overall effects were estimated by the coefficient of each individual NPI in Bayesian inference models. Reductions in $R_t$ were shown as mean, 50%, and 95% prediction intervals. PCR screening showed the joint effect of mass screening and medicine management (generic antipyretics, not specific drugs for COVID-19). Social distancing measures represented the joint effect of stay-at-home order, business premises closure, public transportation closure, gathering restriction, workplace closure, and school closure. **b** Infections simulated by Intervention-SEIR-Vaccination (ISEIRV) model under all real-world NPIs (curves in brown) or in counterfactual scenarios where social distancing measures (SD), facial mask (FM),

contact tracing (CT), PCR screening (PCR), or all NPIs were not implemented, respectively. Mean and 95% confidence intervals (CI, shaded areas) are presented. The brown dashed lines are the total population of cities with outbreaks of each variant. The gap between the simulated curve without each NPI and the red curve represents the effect of each removed NPI in containing the spread. The wider the gap, the higher the effect of NPIs. **c** The ratio of the area under the cumulative infection curve for the corresponding scenario (with one NPI removed) to the area under the baseline scenario curve (with all NPIs removed). The closer it gets to 100% indicates the more effective the removed NPI is for the respective variant.

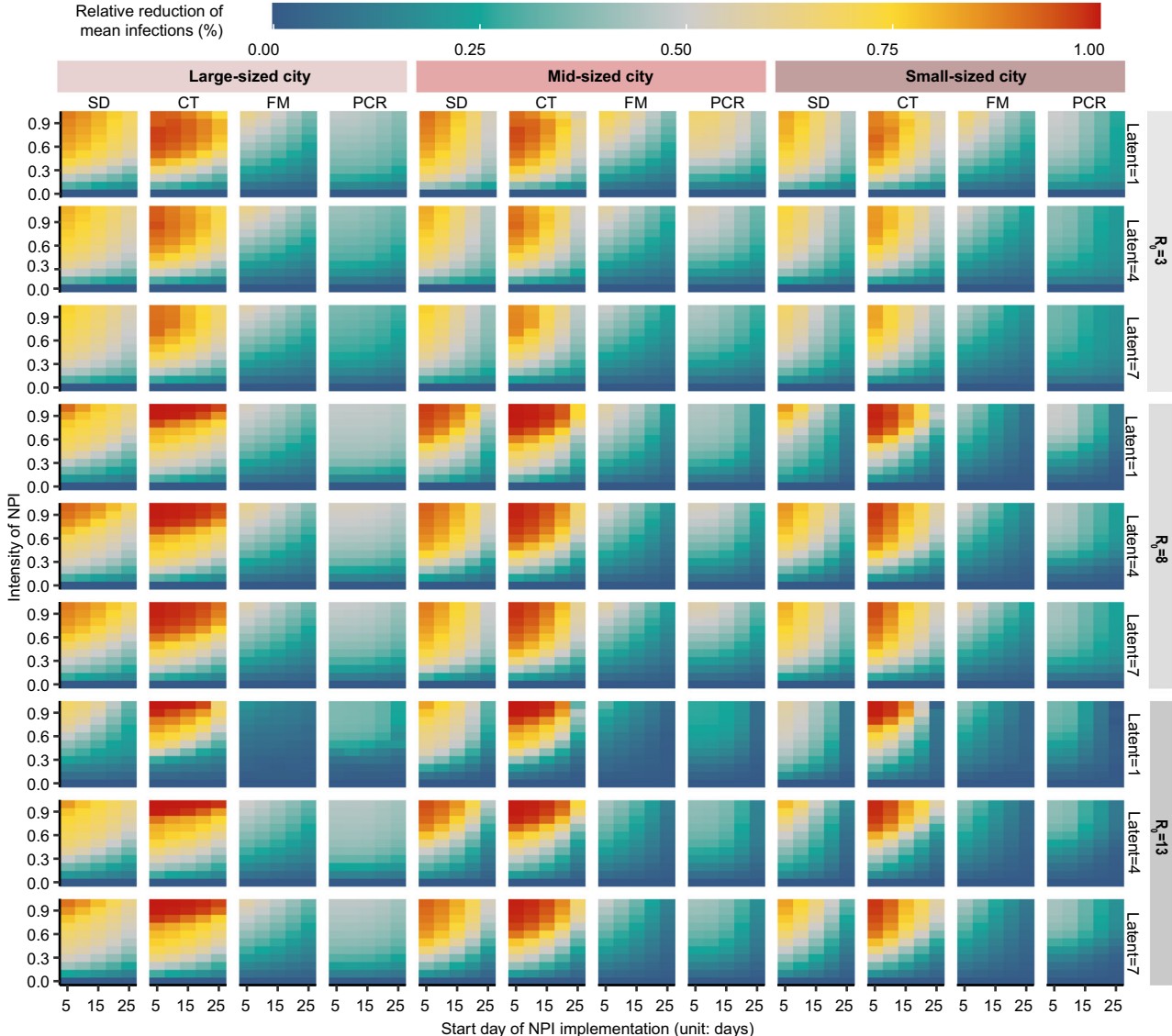

**Fig. 3 | The relative reduction of infections of emerging pathogens under different scenarios of transmissibility, interventions, and population sizes.** Relative reduction of mean infections is the daily mean infections for each scenario, relative to the baseline counterfactual scenario without any interventions. A value closer to 1 (red) indicates fewer infections for a scenario, corresponding to a more effective implementation of NPIs. The values show simulated transmission under different $R_0$ values and latent periods in large cities (LC, over 10 million

residents), medium cities (MC, 5–10 million residents) and small cities (SC, 0–5 million residents). In each small heatmap, the $x$ axis is the start day of NPI implementation for each outbreak and the $y$ axis is the intensity of NPI. The results shown here are the independent effect of social distancing measures (SD), contact tracing (CT), facial masks (FM), and PCR screening (PCR), respectively, meaning that if one NPI was in place, all other interventions were not implemented.

measures were necessary to zero a highly transmissible disease (e.g., infections with an $R_0$ of 13) within 3 months. That means a significant investment of time, effort, and resources that might not be feasible for a long time in the real-world.

## Discussion
Based on China's zero-COVID practice and data in 2020–2022, we evaluated the impact of different public-health interventions for eliminating emerging respiratory viral infections in diverse settings.

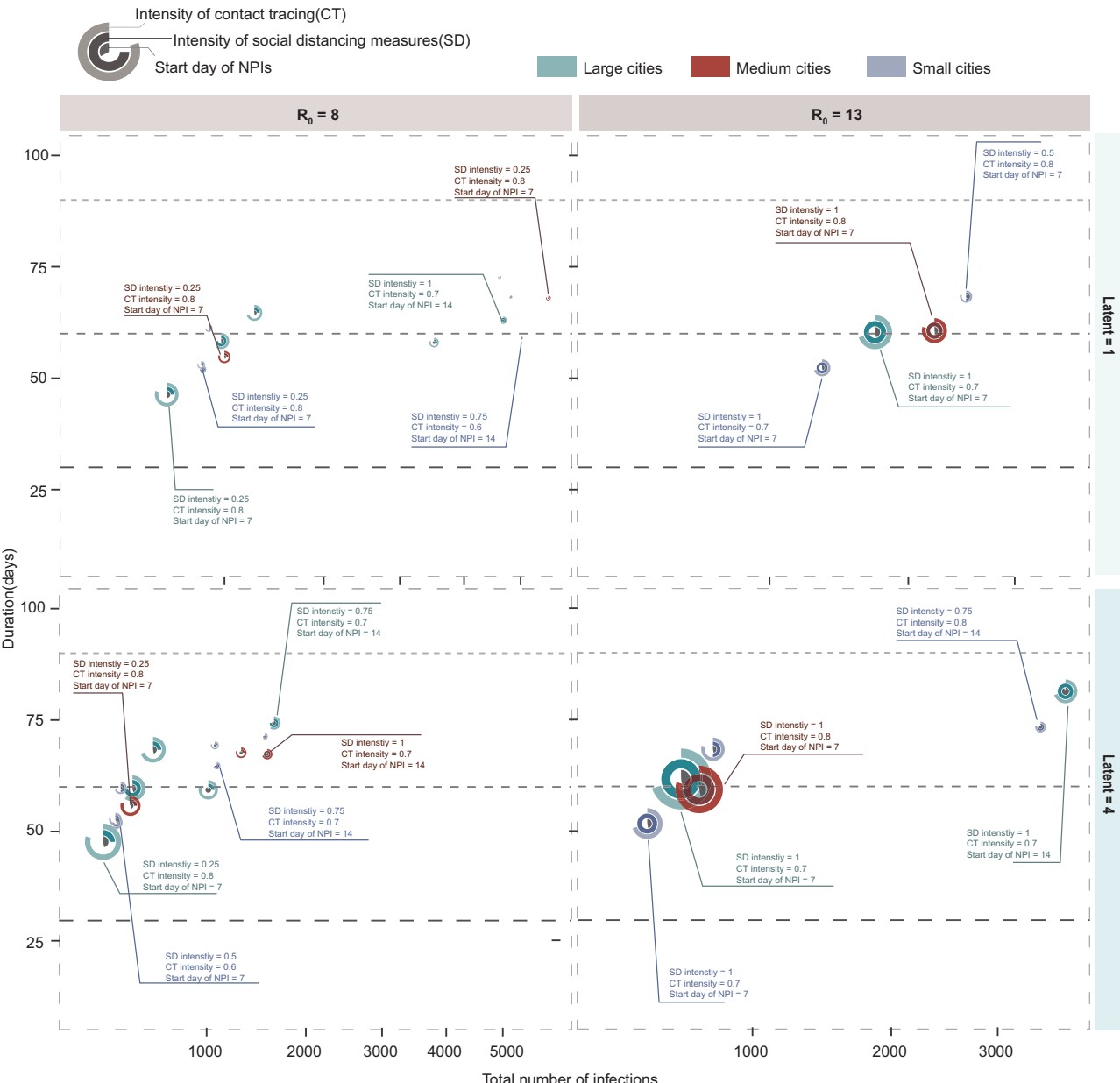

**Fig. 4 | Effectiveness of combined interventions in cities with different population sizes and transmission scenarios.** The *x* axis represents the mean value of the total number of infections in the corresponding size of cities, while the *y* axis represents the mean value of the durations in the corresponding size of cities. The rings of each combination represent the start day of NPI implementation for each outbreak (innermost ring), social distancing intensity (middle ring), and contact tracing intensity (outer ring), respectively. The green, red, and blue colors correspond to large cities (LC), medium cities (MC), and small cities (SC). We considered each combination to be effective when it is valid for more than three cities in parallel. We assumed a 50% probability that an individual wears a mask when in contact with an infected person during outbreaks, considering feasibility and generalizability to the other countries/areas. The simulation results of other scenarios are available in supplementary (See Supplementary Figs. 16–38).

We found that containing highly transmissible infectious diseases with short incubation periods through NPIs can be particularly challenging, due to the relatively heterogeneous effectiveness of interventions under varying epidemiological and socioeconomic contexts. The relative effect of public-health measures on zeroing out transmission was sensitive to the infectivity of pathogens, the timing and intensity of NPIs, and combinations of interventions, which should be considered in tailoring response strategies at different stages of an emerging epidemic or pandemic in various regions[14,15,36]. For instance, similar to findings in the United Kingdom[37] and the United States[38], contact tracing was found to be more effective than social distancing and large-scale tests at the early stages of outbreaks, in curbing transmission by pinpointing the infected individuals and isolating close contacts in communities[39–42], while social distancing represents a general requirement applied to all individuals in an area, regardless of people's infection or exposure status. However, if there are many scattered cases or a significant proportion of the population infected in the middle to late stages of the outbreak, we might expect social distancing and masking to work better than mass screening and contact tracing[37,43,44]. In places with considerable population flow, contact tracing and mass screening might be more effective than social distancing and masking in reducing imported infections[40,45]. Nonetheless, mass testing could be increasingly laborious and time-consuming facing a large population size. Pre-symptomatic or asymptomatic infections might also weaken the impact of population-scale testing, as many infected individuals cannot be identified or the gatherings in test

sites could potentially increase the risk of transmission[46]. Nevertheless, facial masking played a most stable role, especially given its role in reducing the risk of transmission in households or indoor settings[46–50]. In light of this, many governments worldwide have responded to the emergence of COVID-19 by enacting legislation mandating mask usage in public places[51]. Additionally, the continued significance of mask-wearing as a vital tool in East Asia to control the spread of respiratory infections is well-documented[52].

The surge of cases and delayed responses during an outbreak might overwhelm the system of tracing and testing[53], leading to a reduction in the likelihood of cases being properly identified. In such scenarios, social distancing measures could play a more important role for containing outbreaks that have spread widely, showing its stronger cumulative effects in simulations over a longer time period. These measures might be relatively effective, straightforward, and easy to implement[14,54], compared to other interventions that require changes in individual health behaviors (such as mask wearing) or involve the deployment of resources (such as testing kits and laboratories). Moreover, as highlighted by Brauner et al.[30], the combined implementation of certain NPIs has the potential to yield even more optimal effectiveness in mitigating pandemics. However, maintaining the social distancing policy over time could lead to potential adverse consequences, including negative socioeconomic and mental health impacts[55–57]. Governments and health departments, therefore, should remain vigilant and shift their strategies as appropriate.

While the data from the zero-COVID policy provided real-world evidence of interventions for zeroing out outbreaks caused by variants with different severities, however, the lack of knowledge about emerging contagions renders challenges in conceiving precise NPIs packages as well as their efficient implementation. Simulated outbreaks indicated that tracing and isolation alone can control outbreaks caused by less transmissible contagions. Nevertheless, if a disease with high clinical severity spreads rapidly in a population, further social distancing interventions like school closures and gathering restrictions might be necessary as supplements. In addition, as the transmissibility increased, social distancing measures might show a decreasing and lagged impact in reducing the spread in communities[58,59]. We note that rapid growth in the number of cases caused by a highly contagious disease would narrow the control window, a time delay where spread can be contained, and a reminder that the timeliness to implement these universal NPIs is key. However, the high proportion of asymptomatic infections (88%, IQR 41%–100%) observed during the Omicron waves highlights the need for strong organizational proficiency and resource allocation readiness to enable timely interventions and prevent further infections[60–62]. NPIs may therefore have a limited role in containing highly transmissible infectious diseases rapidly, and a mitigation strategy may be the alternative solution[63]. There are several limitations to our study. First, we only focused on the role of NPIs in interrupting the transmission of emerging infectious diseases, which we assumed have serious health consequences in terms of hospitalization or mortality. This study did not explore the feasibility of these zero-infection strategies by assessing their direct or indirect socioeconomic costs and health benefits in varying settings. Second, although national guidelines for COVID-19 responses in China have been issued and modified according to the changing situations of the pandemic and the virus[64], local governments were relatively independent in terms of NPI implementation, rendering challenges in quantifying the intensity and subtle changes of interventions in each study outbreak. Third, our conclusions resulted from the assumption of independent effects of NPIs. However, it is important to recognize that public-health measures with different mechanisms may exhibit synergistic effects, such as mask wearing and social distancing, or vaccination with NPIs[24]. We plan to further explore these synergistic effects in the future.

## Methods

### Data sources and processing

**Epidemiological data.** Data sources for outbreaks included press releases from local government websites or reports from the local disease control and prevention agencies at the province, city/prefecture, and district/county levels, as well as updates from official social media accounts of governments or health departments (supplementary Section A). For each outbreak, we collected the basic information of location, start and end dates, strains, and suspected sources, as well as the daily number of new infections and cases identified among close contacts who had been isolated and quarantined. Based on the case information reported from official sources, we defined the onset of an outbreak as the date when new non-isolated cases started to increase. The end of an outbreak was defined as the date when zero new cases were initially reported, followed by a consecutive period of more than 7 days with no new infections. When an outbreak displayed two or more discernible peaks, with a sustained period of over 5 days between these peaks characterized by zero new cases, we designated the appearance of new cases as the beginning of a new wave. To ensure a sufficient sample size for assessing intervention effects, we ruled out the outbreaks that had <50 cases or lasted less than 7 days. Ultimately, a total of 131 outbreaks were employed in the following analyses.

**Public-health measures.** For each outbreak, we collected the corresponding public-health measures from the websites of local governments. The collected public-health measures include: (1) stay-at-home order (SO), (2) business premises closures (BPC), (3) public transportation closures (PTC), (4) gathering restrictions (GR), (5) workplace closures (WC), (6) school closures (SC), (7) facial masking (FM), (8) mass PCR screening (MS), (9) medicine management (MM), and 10) contact tracing (CT). Nine of the indicators are recorded on an ordinal scale representing the level of policy stringency. In China, both MS and CT were deemed essential approaches for infection detection and tracing but differ in definition and implementation as follows. Mass screening was typically conducted for individuals without a clear exposure history but who were at risk of exposure, such as those residing in the same community as a confirmed case. Only if a positive test result was obtained, the individual would be quarantined, and until then they were allowed to move around. However, close contact tracing refers to the detection and isolation of individuals with a history of exposure to a case. These individuals were regularly tested for COVID-19 nucleic acid during the quarantine period. More details on data processing and synthesis can be found in the supplementary Section A.

**Control variables.** The trajectory of infectious diseases can also be influenced by some confounders like weather, season, and vaccination via the effect on viral activity and human behavior. Confounders considered in this study were daily temperature, humidity, population density, and vaccination rate. We derived the daily mean temperature and humidity within each city with data from Google Earth Engine (GEE) from Global Land Data Assimilation System Version 2 (GLDAS 2.1), a dataset with a combination of model and satellite- and ground-based observation data[65]. However, we excluded humidity as a control variable in our model, due to its high collinearity with air temperature. Population density obtained from http://www.stats.gov.cn/. Vaccination data at the province level were collected from http://www.nhc.gov.cn/. Although inactivated vaccines seem to have low efficacy to prevent Omicron infections among the population[66,67], vaccination data were processed as a practical vaccination rate, based on the full vaccination rates and the efficacy of COVID-19 vaccines (see Supplementary Table 2). Please see supplementary Section A.3 for more details on vaccine data processing.

**Ethical approval.** Ethical clearance for collecting and using secondary data in this study was granted by the institutional review board of the

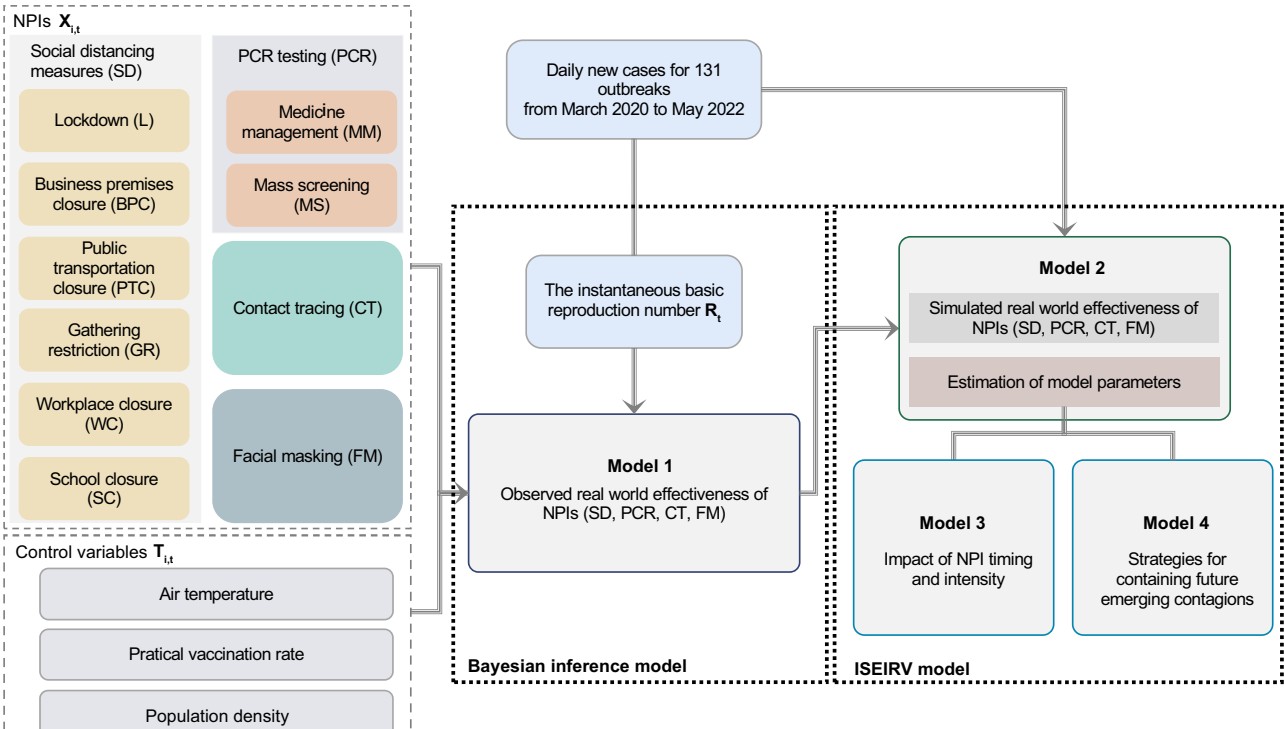

**Fig. 5 | Schematic flowchart of data and models for this study.** A prior on the basic reproduction number $R_{0,v}$ was used for each outbreak, with a hyperprior varying by SARS-CoV-2 lineages (see Supplementary Table 5). Then, we estimated the instantaneous reproduction number ($R_t$) based on the observed daily new cases. By comparing observed $R_t$ with $R_{0,v}$ in a Bayesian inference model, we estimated the coefficients of variables to assess their effects on curbing COVID-19. Ten NPIs were divided into four categories: social distancing measures (yellow),

polymerase chain reaction screening (red), contact tracing (green), and facial masking (blue). Finally, an Infectious-Intervention-SEIR-Vaccination (ISEIRV) model was built to simulate the timing and intensity of NPI implementation and elimination strategies under diverse transmission scenarios. The prior information for parameter estimation within the ISEIRV model was informed by the effectiveness of each NPI category.

University of Southampton (No. 61865). All data were supplied and analyzed in an anonymous format, without access to personal identifying information.

**Bayesian inference model**

We built a Bayesian model to infer the effects of NPIs on the reduction in instantaneous reproduction number ($R_t$) (see Fig. 5). In this study, $R_t$ was estimated by a Bayesian framework[68] (see supplementary section B for more details). For a robust, localized estimation, we assumed that the effectiveness of the individual NPI was relatively stable in each outbreak regarding the same variant and location. The effectiveness of each NPI was evaluated by estimating how much of the variation in NPI intensity could account for the reduction in $R_t$. Specifically, a generalized linear relationship between NPI intensity and the reduction was considered below. The relationship between each individual NPI and effect on the reduction of $R_t$ was assumed to be linear. This assumption was incorporated into a hierarchical model, where varying slopes were used for each variant. Non-informative priors with a Gamma distribution were included in the model consistent with methodology employed by Yong et al. and others[20,24,30,69].

$$\Phi_{t,v} = R_{0,v} \prod_{i=0}^{n} \exp(-\alpha_{i,v}X_{i,t,v} - \beta_i T_{i,t,v}) + \varepsilon 1 \quad (1)$$

$$\Phi_{t,v} \sim gamma(R_{t,v}, \sigma_{t,v}) \quad (2)$$

Where $X_{i,t,v}$ indicates each individual NPI $i$ under variant $v$ on day $t$. n represents the number of NPIs. $T_{i,t,v}$ indicates the control variable, i.e., air temperature, population density and practical vaccination rate used in this study. $\alpha_{i,v}$ represents the coefficient of the covariate $X$

across different variants v. Following the assumption by Flaxman et al.[14], the NPI coefficients are assigned a Gamma distribution with shape parameter 1/6 and scale parameter 1 and further shifted by log (1.05)/6 to accommodate for both positive and negative effects, see supplementary Fig 5. $\beta_i$ represents the coefficient of the control variable. We placed it a Normal prior with a coefficient of $N(0, 0.5)$, indicating $\beta_i$ might also be associated with the trajectory but remains constant across all variants. The contribution of other unobserved confounding factors to the $R_t$ decay is indicated by the residual $\varepsilon$. We set the prior of the basic reproduction number $R_{0,v}$ to obey a Gamma distribution with a shape parameter of $f$ and a rate parameter of 0.1, where f varied with SARS-CoV-2 lineages (see supplementary Table 5). $\sigma_{t,v}$ indicates the variance of $R_{t,v}$, which was estimated by the observed cases data on a daily basis for each study outbreak. Note that each NPI has been normalized to 0−1. The details of pre-processing NPIs can be found in supplementary method A.2 and supplementary Equation Table (https://github.com/wxl1379457192/Zeroing_out_emerging_contagions/blob/main/Supplementary_equation_table.doc).

Finally, we estimated the effectiveness of each NPI group under each variant using Markov Chain Monte Carlo methods with Rstan[70]. We ran 5 parallel chains for 10,000 iterations with 5000 iterations for warmup and a thinning factor of two to obtain 12,500 posterior samples. And the effect of NPIs can be computed by $1 - \exp(-\sum_i^n \alpha_{i,v} \overline{X_{i,t,v}})$, which $\overline{X_{i,t,v}}$ represents the median intensity of each individual NPI $X_i$ for variant $v$. n represents the number of NPIs in each category. The median value of NPI intensity was calculated based on the normalized NPI. The NPIs effect was defined as the percentage reduction of $R_t$ relative to $R_0$ with its highest value of 1, representing the outbreak has been fully contained. The leave-one-out cross validation method was used to validate our model

(see supplementary Section B). We also performed a sensitivity analysis to assess the robustness of model parameter assumptions (see supplementary Section E).

### Reconstructing and simulating transmission under varying scenarios

**Compartmental model.** The classic SEIR model divides the whole population into four components, including the susceptible ($S$) population which can be infected, the exposed ($E$) population which has been infected but not yet be infectious, the infectious ($I$) population who can infect the susceptible, and the recovered or removed ($R$) population which cannot be infected or spread pathogens to other people. We modified the classic SEIR model into Intervention-SEIR-vaccination model (ISEIRV) to account for the impact of NPIs implementation and large-scale vaccination:

$$N = S(0) + E(0) + I(0) + R(0) \tag{3}$$

$$S(t+1) = S(t) - \frac{\beta(t)(1 - c(t))I(t)S(t)}{N} - v(t)S(t) \tag{4}$$

$$E(t+1) = E(t) + \frac{\beta(t)(1 - c(t))I(t)S(t)}{N} - \lambda E(t) \tag{5}$$

$$I(t+1) = I(t) + \lambda E(t) - r(t)I(t) \tag{6}$$

$$R(t+1) = R(t) + r(t)I(t) + v(t)S(t) \tag{7}$$

As vaccines have been used to mitigate COVID-19 transmission in the middle of our study time period, we added an additional term $v(t)S(t)$ in the dynamics of the susceptible population, where $v(t)$ is the daily practical vaccination rate transferring the susceptible population to the virtually recovered population with vaccine-introduced immunity. Given the short period of simulations for each outbreak, the overall low efficacy of vaccines to prevent infections, and the fact that the majority of populations have not been infected, the decline in immunity during the outbreaks was not considered. $\lambda$ is the inverse of the latent period representing the transfer rate from the exposed population to the infectious population, i.e., the time lag between being infected and being infectious.

The transmission rate $\beta$ between the susceptible and the infectious populations was designed as a function of time to reflect the impact of NPI implementation. That is,

$$\beta(t) = b_0 \exp(-b_1 x_1(t))*(1 - 0.25 x_2(t)) \tag{8}$$

where $b_0$ is the baseline of the transmission rate without any interventions, $x_1$ is the integrated measure of the strengths for the contact reduction aimed NPIs, and $x_2$ is the fraction of the population that complied with the requirement to wear a mask. Efficiency of facial masking for preventing indoor transmission was set as 25%[71]. The general measure of contact reduction was calculated by NPIs including (1) stay-at-home order, (2) business premises closure, (3) public transportation closure, (4) gathering restriction, (5) workplace closure, and (6) school closure. The intensity of the general contact-reduction measure was then defined by the linear combination of the intensities of relevant NPIs on a daily basis:

$$x_1 = \sum_{i=1}^{6} w_i * X_i \tag{9}$$

$$w_i = \frac{e_i}{\sum_{i=1}^{6} e_i} \tag{10}$$

where the weights $w_i$ for each NPI $X_i$ was proportionally determined by its empirical $e_i$ of reducing $R_t$ assessed by the Bayesian inference model. Next, the dynamic transmission rate was further adjusted by the contact tracing rate $c(t)$, which measures the daily ratio of cases identified among close contacts in isolation to all cases reported.

The detection and quarantine of infectious populations could reduce the probability of transmission. Therefore, we also modified the recovery/removed rate $r$, to account for the implementation of mass testing, which moves infectious people into the compartment R. In specific, $r$ was modeled as

$$r(t) = \frac{1/(r_0 \exp(-r_1 x_3(t)))}{1 + \exp(-r_2(t - r_3))} 11$$

where $r_0$ is the infectious period for omicron[72], and $x_3$ is the intensity of mass screening. For pre-Delta and Delta periods, we adjusted $r_0$ as 7 and 6, respectively. The term, $1/(r_0 \exp(-r_1 x_3(t)))$, indicates the expectation of the virtual recovery rate. Considering the policy lag and process of large-scale mass screening, the recovery rate was adjusted in the shape of logistic function by dividing $1 + \exp(-r_2(t - r_3))$. $r_1, r_2$ and $r_3$ are underestimated model coefficients. More details on parameter calibration and model validation for ISEIRV can be found in the supplementary method.

**Scenario analysis of zeroing strategies.** To reveal the effects and challenges of NPIs and intervention timings on zeroing out emerging contagions under different scenarios, we further simulate the spread of emerging respiratory viral infections in cities with varying population sizes (<5 million, 5–10 million, and over 10 million residents) (See supplementary Table 7). We simulated the epidemiological trajectory of the spread of the virus within each city using ISEIRV model in MATLAB version R2022a, under 1800 scenarios (4 NPIs × 10 intensities × 5 time-points × 9 epidemiological parameter schemes) including nine combined transmission settings (set $R_0 = 3, 8, 13$, and incubation period ($I_t$) = 1, 4, 7 days, respectively). The simulations focus on the time-point and the intensity to implement NPIs, including social distancing measures (SD), facial masking (FM), PCR screening (PCR), and Contact tracing (CT). Effects of the implementation of independent and combined NPI were also assessed (see our supplementary Section D for details). For each transmission parameterization scenario, we simulated infection curves with zero-implementation as benchmark.

### Reporting summary
Further information on research design is available in the Nature Portfolio Reporting Summary linked to this article.

## Data availability
All the data used in this study are publicly available online at: https:// github.com/wxl1379457192/Zeroing_out_emerging_contagions.

## Code availability
The modeling codes for this study are available online at Zenodo: https://doi.org/10.5281/zenodo.8195369. The code for processing climate data on GEE (Google Earth Engine) can be found via https://code. earthengine.google.com/81f7cd2ef122d3f7f597a63cc603196d.

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

## Acknowledgements

We thank the researchers and organizations who generated and publicly shared the mobility, epidemiological, intervention, and analyzing code used in this research. This study was supported by the National Institute for Health (MIDAS Mobility R01AI160780), the Bill & Melinda Gates Foundation (INV-024911), the National Nature Science Foundation of China (42230110, 72025405, 91846301, 72088101, 42222110), and the Beijing Natural Science Foundation (7202073). The funders of the study had no role in study design, data collection, data analysis, data interpretation, or writing of the report. The corresponding authors had full access to all the data in the study and had final responsibility for the decision to submit for publication. The views expressed in this article are those of the authors and do not represent any official policy.

## Author contributions

S.J.L. and Y.G. conceived and designed the study, interpreted the findings, and wrote the manuscript. X.L. Wu and W.B.Z. built the model, collected data, finalized the analysis, interpreted the findings, and wrote the manuscript. X.L. Wang collected data, interpreted the findings, commented on, and revised drafts of the manuscript. D.Z., Z.P.R. and Y.C.Y. collected data. J.H.W. and H.Y.L. interpreted the findings, and revised drafts of the manuscript. N.W.R., C.W.R., E.C., A.W., D.A.T.C., Z.J.L. and A.J.T. interpreted the findings, and commented on and revised drafts of the manuscript. All authors read and approved the final manuscript.

## Competing interests

The authors declare no competing interests.
