## [Peer Review File · Nature Communications]

Effects of public-health measures for zeroing out different SARS-CoV-2 variantsREVIEWER COMMENTS

Reviewer #1 (Remarks to the Author):

Dear authors,

I am thankful for the opportunity to review your study estimating the effects of public health measures (social distancing measures, facial masking, screening, and contact tracing) in containing more than 131 local outbreaks in China between 2020 and 2022, thereby covering multiple SARS-CoV-2 variants.

Overall, I am impressed by the collected data and the effort that went into the analyses. I think it is also very interesting to compare the effects of public health measures over time for different SARS-CoV-2 variants. I am a bit skeptical about the impact of the findings because numerous studies have already estimated the effects of public health measures and it seems to me that this work does not provide many new, surprising findings. Maybe the editors and other reviewers can also comment on this. I will subsequently focus on the quality of the statistical analysis.

The analysis can be separated into three steps: 1) Estimating the analyzed outcome (the reproduction number) from the observed outcome (cases); 2) Estimating the effects of public health measures; 3) Using the empirical estimates to run different scenarios to understand the influence of intensity, timing, and combination of public health measures. My review will focus mostly on 2) - my own area of research.

1) Estimating the reproduction number: The EpiEstim framework is commonly used to estimate the time-varying reproduction number. I suggest having a look at EpiNow2 by Abbott et al. 2020 [1], which is based on EpiEstim but includes an observational model that allows considering further time lags from infection to reporting.

I would encourage you to plot the raw data, i.e. the number of new cases over time for each outbreak. Furthermore, I couldn't find a definition of the start and end of an outbreak. How were they defined and were they defined consistently across outbreaks? Start and end date can influence the estimation of R_t and the estimated effects of public health measures if the dates are set too early or too late.

Steps 1-3) entail a lot of assumptions and priors, but the Supplements barely contain graphical displays of the corresponding distributions of priors. For example, the serial interval of each SARS-CoV-2 variant (Supplementary Table 3) could be plotted and compared in one figure. Similarly, I suggest adding displays for other delay distributions and important priors such as the effect of non-pharmaceutical interventions (NPIs). Graphical displays will make it easier for readers to assess the assumptions and follow the methods.

The estimation of R_t is separated from the empirical estimation of the effects of public health measures, thereby ignoring uncertainty from step 1) in step 2). Separating step 1) and 2) is fine, but you should somehow incorporate uncertainty from step 1) in 2). Alternatively, you could perform 1) and 2) in a single model like others [2-3].

2) Estimating the effects of public health measures: I like that the authors collect a comprehensive dataset of public health measures that considers varying intensity of the measures. I am not sure though if it is good practice to normalize the ordinal NPIs into a variable ranging from 0 to 1. Most studies consider NPIs as binary indicator variables [4]. I think you could do the same by encoding each ordinal variable as multiple binary variables. It depends on how much variation you have between outbreaks whether you can estimate the effects of each level of an NPI. If not, you may use a less granular coding of NPIs.

I don't think that references 13 to 16 constitute representative examples of previous studies on the effects of public health measures. For one thing, Ref 13 presents data rather than analysis and Ref 16 is a Matters Arising concerning the study of Flaxman et al. [2] and should probably be cited together. I think that references 59 (Brauner et al.) and 61 (Haug et al.) would be more prominent examples of this stream of literature. You may further draw from the list of studies reviewed by Banholzer et al. [4] for possible additions to your related work.

It is very interesting to compare the effects of public health measures between different SARS-CoV-2 variants. To my knowledge, this is quite unique and it is also important to understand how the effectiveness of measures depends on the variant, or more generally, the time and context in which they were implemented. I have only a minor suggestion here. Instead of splitting the data into groups by variant, the authors could analyze them jointly with a hierarchical model using varying slopes on the effects of public health measures for each variant. Thereby, information across outbreaks for the estimation of all non-variant specific parameters are shared, which should be more efficient for model estimation.

The reporting of the effects of public health measures with median and IQR is rather unusual. I think the vast majority of studies report mean/median and 95% credible interval. I can see that a leave-one-out cross validation was performed in the Supplements, thus you could also report prediction intervals as in Brauner et al. [3], which reflects even greater uncertainty in the estimates.

The formula on page 19 line 388 probably has one "exp" too much.

In Supplementary Text A.2, I did not quite understand what exactly a_j is (the size of the geographic scope). Is it the size of the population? If you want to consider that only a subpopulation was affected by the measure, then you could weight the effects of public health measure by the proportion of the population that is affected, as e.g. in Banholzer et al. [5].

The prior for the effects of social distancing measures could be borrowed from Brauner et al. [3]. They use an asymmetric prior where positive effects (reduction in R_t) are more probable, but the prior still allows for negative effects (increase in R_t).

I think the third and fourth limitation of the study should be placed more prominently before or in the Result section. It is important context regarding the magnitude of the effects that they are conditional on the already implemented long-lasting measures. It is also important context that your exclusion criteria for outbreaks most likely underestimates the effectiveness of contact tracing or other measures that managed to control the outbreak quickly.

3) Scenario modeling: I cannot comment much on this step as it is not directly my area of expertise. I was a bit puzzled by the statement in line 172 that the results from the ISEIRV model validate the results from the Bayesian inference model. Anything else would surprise me given that the parameters for the effects of public health measures of the ISEIRV model are informed by the Bayesian inference model.

I commend the authors for the time and effort invested in this study and wish them best of luck with any revisions. I hope my comments are helpful for this.

Kind regards,
Nicolas Banholzer

References:

[1] Abbott S, Hellewell J, Sherratt K, Gostic K, Hickson J, Badr HS, DeWitt M, Thompson R, Funk S. EpiNow2: estimate real-time case counts and time-varying epidemiological parameters. R package

version 0.1. 0. 2020.

[2] Flaxman S, Mishra S, Gandy A, Unwin HJ, Mellan TA, Coupland H, Whittaker C, Zhu H, Berah T, Eaton JW, Monod M. Estimating the effects of non-pharmaceutical interventions on COVID-19 in Europe. *Nature*. 2020;584(7820):257-61.

[3] Brauner JM, Mindermann S, Sharma M, Johnston D, Salvatier J, Gavenčiak T, Stephenson AB, Leech G, Altman G, Mikulik V, Norman AJ. Inferring the effectiveness of government interventions against COVID-19. *Science*. 2021;371(6531):eabd9338.

[4] Banholzer N, Lison A, Özcelik D, Stadler T, Feuerriegel S, Vach W. The methodologies to assess the effectiveness of non-pharmaceutical interventions during COVID-19: a systematic review. *European Journal of Epidemiology*. 2022;37(10):1003-24.

[5] Banholzer N, Van Weenen E, Lison A, Cenedese A, Seeliger A, Kratzwald B, Tschernutter D, Salles JP, Bottrighi P, Lehtinen S, Feuerriegel S. Estimating the effects of non-pharmaceutical interventions on the number of new infections with COVID-19 during the first epidemic wave. *PLoS one*. 2021;16(6):e0252827.

Reviewer #2 (Remarks to the Author):

General comments:

1. Many thanks to the authors for an interesting set of analyses.
2. I suggest adding some more specificity in the descriptions of the NPIs in the main paper, particularly the mass screening and contact tracing scenarios. The supplemental information provides much more detail (particularly in lines 30-44 in the supplement). While the terms "mass screening" and "contact tracing" are familiar in general, I think it would be helpful for readers to see the specific descriptions associated with China's implementation(s) of the zero-COVID policy (e.g., the implementation of contact tracing isn't defined until lines 250-251. This is particularly relevant for mass screening which was implemented in very different ways at different times in different countries.
3. In addition to the definitions requested in the previous general comment, I also suggest adding some brief discussion of the role of mass screening and contact tracing NPIs compared to social distancing and masking. Social distancing and masking are general requirements applied to all individuals in an area, regardless of their own infection or exposure status while contact tracing and mass screening focus on isolating detected cases or exposed individuals respectively. This difference between NPIs applied to all or NPIs designed to detect individuals who will then be subject to isolation seems important in the discussion of effectiveness early or late in an emerging pandemic. If there are many scattered cases, we might expect SD and M to work better than MS and CT, which is what was observed.
- 4.

Specific comments:

1. Lines 43-45 (and throughout). The effects are described as percentages without specifying that these are percent reductions in the instantaneous reproductive number. While it may seem repetitive, I suggest clarifying that the effect is "a 23% reduction in the instantaneous reproductive number" on line 43, and "xx% reduction" in the following sentences. Currently the effects are described without context or interpretation.
2. Line 45. Omit "However".
3. Line 58. "so as to avoid developing". Should this be "so as to avoid the epidemic developing"?
4. Line 71. I suggest replacing "such as" to "including". Mental health is often overlooked among lists of secondary health effects so I suggest "including" it rather than listing it as an example.
5. Line 128. I suggest referencing supplemental figure 1 (the map) here as well.
6. Line 129. "intensity of lockdown". I suggest adding more detailed description of the (0,1) assessment of intensity in the main text prior to its use here. The parenthetical definition in line 131-133 is fairly vague and details in the supplemental information are not referenced here. This could be accomplished with a brief (2-3 sentence) paragraph setting the stage.
7. Line 206. I suggest omitting "relatively".

8. Line 273. What does "facial masking is set to 0.5" mean in terms of implementation? It is the middle of the (0,1) range but since masks were mandatory, I'm not sure if this refers to compliance? I suspect it does not mean only 50% of individuals were wearing masks, but it would be helpful to calibrate this mid-level assessment in light of the entire zero-COVID policy.
9. Line 287. "The prompt" to "Prompt".
10. Line 308. "The data of" to "The data from".
11. Line 344. "Data sources of" to "Data sources for".
12. Line 355. "the local Centers for Disease Control and Prevention". Does this refer to Province-level offices? This is not clear to me.
13. Line 386 and following. The relationships have linear components but most are transformed so I suggest referring to "generalized linear" relationships throughout.
14. Line 395. "the coefficients are placed as a Gamma prior with hyperprior, following $\text{gamma}(u,1)$ and $u \sim \text{uniform}(0,1)$." I suggest "the coefficients are assigned $\text{gamma}(u,1)$ prior distributions and u assigned a $\text{uniform}(0,1)$ hyperprior distribution". Also, a gamma prior on association parameters would only allow positive estimates of association parameters, is this what is intended?
15. Line 410. Is the underlining of $X_{i,t}$ intentional? Or is there supposed to be a bar on $X_{i,t}$ in both places to indicate the average?
16. Line 412. "normalized NPI". I assume this refers to the (0,1) scoring of the NPIs but the reference to normalization is not clear until deep into the supplemental information. If a brief description and interpretation of the (0,1) range of values for NPI is added earlier on (as suggested in a few comments above), there would not be a need for the "normalized" qualifier.
17. Line 419. "Divided" to "divides".
18. Line 446. "ratio of facial masking". As in other comments, clearly defining the NPI scores used would be helping. Here "ratio of facial masking" does not refer to the fraction masking. Also, how is "efficiency of facial masking for preventing indoor transmission" defined and implemented in the model? I don't believe the authors' model differentiates between indoor and outdoor transmission, does it?
19. Line 451. "were linearly" to "was linearly".
20. Line 454. "the weights...was" to "the weights...were". Also, the phrase "the weights were determined by their effects". What do the authors mean by "effects"? It would seem circular to weight based on the estimated effects but I feel I'm missing something.
21. Line 465. "recovery rate then" to "recovery rate".
22. Line 498. What does "GEE" refer to here?
23. Supplement, line 56. What do the parantheses denote here?
24. Supplement. The Bayesian inference model description is presented across a long narrative, I would find it helpful to have equations for all of the elements (equations, priors, hyperpriors) in a Box to present along with the narrative so all of the elements are in one place and the model definition is separated from details of its implementation. I feel this would provide a good context for the listing of the sensitivity analysis priors on lines Supplement 413-418.
25. Supplement, lines 299-307. Is it fair to refer to this approach as approximate Bayes' computation (ABC)? Parameters generated from priors, model generates output that updates the prior to yield the posterior?
26. Supplement, line 311. "splitted" to "split".
27. Supplement, line 329. The adjusted R-squares above 99% seem remarkably high. Is this common in model validation?
28. Supplement, lines 392 and following. The stategy comparsion visualizations are quite interesting, thank you for sharing (they may well merit a publication on their own).

Reviewer #3 (Remarks to the Author):

This is an interesting study that gathered detailed COVID-19 and NPI data for 131 outbreaks occurred from April 2020 to May 2022 in China. While many previous works examined the effects of NPIs on

COVID-19 in different countries and settings, this study is unique given the zero-COVID policies strictly enforced during the study period in China. The study first estimated the effects of ten interventions using a Bayesian inference framework and then used a compartmental model to simulate the effects of NPI combinations under different scenarios. Findings were effectively communicated using customized visualization tools (e.g., Fig. 4). Overall, I think this is a significant contribution to the literature on quantifying NPI effects on infectious disease outbreaks, which provides new insights in the context of eliminating emerging and re-emerging pathogens. I have a few comments that hopefully can help further improve the manuscript.

1. The study used an intensity level for interventions defined based on a set of criteria. For a better context, it would be useful to have some examples in the main text to show the full spectrum of intervention intensities. For instance, what does an intensity of 0.3 mean?

2. Line 176. The Bayesian inference revealed some synergistic effects between interventions. It seems the effects of NPIs were assumed independent and multiplicative on R_0 (see the equation in line 388). Could the authors elaborate more on how synergistic effects were quantified?

3. Small-scale outbreaks are highly stochastic, especially for emerging contagions at the early phase. I am wondering if simulations using the ISEIRV model considered such dynamical stochasticity and if this will impact the results. Maybe the authors can integrate the compartmental model stochastically using some distributions for new infections.

4. It would be good to discuss the generalizability of the findings in other countries. Particularly, the feasibility and challenges associated with those NPIs.

5. A typo in line 388, an extra "exp" in the equation.

6. Here is a relevant study that may be useful: Peak, C.M., Childs, L.M., Grad, Y.H. and Buckee, C.O., 2017. Comparing nonpharmaceutical interventions for containing emerging epidemics. *PNAS*, 114(15), pp.4023-4028.

Reviewer #1 (Remarks to the Author):

Dear authors,

I am thankful for the opportunity to review your study estimating the effects of public health measures (social distancing measures, facial masking, screening, and contact tracing) in containing more than 131 local outbreaks in China between 2020 and 2022, thereby covering multiple SARS-CoV-2 variants.

Overall, I am impressed by the collected data and the effort that went into the analyses. I think it is also very interesting to compare the effects of public health measures over time for different SARS-CoV-2 variants. I am a bit skeptical about the impact of the findings because numerous studies have already estimated the effects of public health measures and it seems to me that this work does not provide many new, surprising findings. Maybe the editors and other reviewers can also comment on this. I will subsequently focus on the quality of the statistical analysis.

Response: The most interesting part of this study lies on the validation of the estimated effects of NPIs under the varying transmissibility of variants of concern (VoCs), including Alpha, Delta, and Omicron. The existing studies on NPI effects mainly focus on the original virus, SARS-CoV-2, and usually under a relatively stable context of VoCs. Considering that zero-Covid policy is quite unique in its long-time dynamic implementation of NPIs against different variants, we collected the related data in China which can help us to understand how the effectiveness of measures depends on the variant, or more generally, the time and context in which they were implemented. Such new knowledge can also help us to conceive a smart policy package for the future emerging contagion before there is access to pharmaceutical interventions. Besides, the effect of mass screening has been little studied before. Therefore, we believe this study is still quite innovative and should be of interest to readers.

The analysis can be separated into three steps: 1) Estimating the analyzed outcome (the reproduction number) from the observed outcome (cases); 2) Estimating the effects of public health measures; 3) Using the empirical estimates to run different scenarios to understand the influence of intensity, timing, and combination of public health measures. My review will focus mostly on 2) - my own area of research.

1) Estimating the reproduction number: The EpiEstim framework is commonly used to estimate the time-varying reproduction number. I suggest having a look at EpiNow2 by Abbott et al. 2020 [1], which is based on EpiEstim but includes an observational model that allows considering further time lags from infection to reporting.

Response: Thanks for the suggestion. In fact, the time lags from infection to reporting have already been considered in our previous analyses. Nonetheless, as you suggested, we also estimated the time-varying reproduction number (R_t) using EpiNow2 by Abbott et al. 2020 [1]. Then, we compared our previous estimates (R_t from EpiEstim with adjustment of exposure-to-reporting lags) with those of EpiNow2, see Figure 1. The parameters used in EpiNow2, including the generation interval and incubation period, are presented in Table 1. Additional comparative results are available in the attached document "Rt_estimation_based_on_EpiEstim&EpiNow2.pdf" via link https://github.com/wxl1379457192/Zeroing_out_emerging_contagions/blob/main/Rt_estimation_based_on_EpiEstim%26EpiNow2.pdf. In general, the trends of the results of EpiEstim and EpiNow2 are highly similar in most outbreaks.

However, EpiNow2 only presented R_t estimates for the period with case reported (from the first reporting date to the last date) while our approach could directly generate a longer period of R_t estimates starting from the inferred infection date of cases) in the study outbreaks, allowing us to evaluate the impact of NPIs on transmission before case numbers were official released and also possibly making it easier to align the dates between NPI implementation and infections across the outbreak. Therefore, it sounds reasonable to retain our original approach, given the specific needs of our NPI impact analyses.

The adjustment of time lags from infection to reporting has been emphasized in the Method, see line 183-200. More details can be found in SI, Section B.1. For your easy reference, we pasted the related content here: *"...first, we adjusted the lag from exposure to reporting, to account for the incubation period (i.e., time lag from infection to illness onset or the first positive test) and the reporting delay (i.e. time lag from illness onset or the first positive test to reporting). Specifically, for each case, we inferred a time-lag t' , resulting in the infection occurring $t - t'$ days before being reported on day t . This time lag t' is the sum of the incubation period (t'_i) and the reporting delay (t'_r). The incubation period for each case was sampled from a log-normal distribution with varying means and standard deviations for four main SARS-CoV-2 variants (supplementary Table 4 and Fig 4). The reporting delay was also sampled from a log-normal distribution with a mean of 0.82 days and standard deviation of 0.84 days (see supplementary Fig 5). Recognizing that large-scale screening following the reporting of index cases might shorten the reporting delay, after the index case was reported, we re-parameterized the onset-to-reporting lag according to a binomial distribution with a mean of 1 day and a standard deviation of 0.3 days (see supplementary Fig 5). The total number of infections on any given day was then counted by aggregating cases by day after adjusting their exposure-to-report delays. We repeated this random sampling process 50 times to get more robustness results. The daily number of infections was calculated as the daily mean value of the 50 samples."*

Table 1. Generation time and incubation period of each variant used in EpiNow2.

Generation time			
Strains	Mean in days	Standard deviation	References
Original	5.70	3.80	Lau, Yiu Chung, et al. "Joint estimation of generation time and incubation period for coronavirus disease 2019." The Journal of Infectious Diseases 224.10 (2021): 1664-1671.
Alpha	4.70	3.30	Hart, William S., et al. "Generation time of the alpha and delta SARS-CoV-2 variants: an epidemiological analysis." The Lancet Infectious Diseases 22.5 (2022): 603-610.

Delta	2.90	3.00	Zhang, M. et al. Transmission Dynamics of an Outbreak of the COVID-19 Delta Variant B.1.617.2 — Guangdong Province, China, May–June 2021. China CDC Wkly 3 , 584–586 (2021).
Omicron	2.36	0.59	Mesfin, Yonatan, et al. "Epidemiology of infections with SARS-CoV-2 Omicron BA. 2 variant in Hong Kong, January-March 2022." medRxiv (2022): 2022-04.
Incubation period: $t_c' \sim \text{lognormal}(x \mu, \sigma)$			
Strains	μ	σ	References
Original	1.78	0.52	Paul, S. & Lorin, E. Distribution of incubation periods of COVID-19 in the Canadian context. Sci Rep 11 , 12569 (2021).
Alpha	1.50	0.46	Tanaka, H. et al. Shorter Incubation Period among COVID-19 Cases with the BA.1 Omicron Variant. Int J Environ Res Public Health 19 , 6330 (2022).
Delta	1.25	0.34	Ogata, T., Tanaka, H., Irie, F., Hirayama, A. & Takahashi, Y. Shorter Incubation Period among Unvaccinated Delta Variant Coronavirus Disease 2019 Patients in Japan. Int J Environ Res Public Health 19 , 1127 (2022).
Omicron	1.02	0.45	Tanaka, H. et al. Shorter Incubation Period among COVID-19 Cases with the BA.1 Omicron Variant. Int J Environ Res Public Health 19 , 6330 (2022).

Figure 1. Examples of estimated R_t using EpiEstim and EpiNow2 for each outbreak, respectively. Date of infection was derived by removing the infection-to-report time lag from the reported dates. Shaded areas represent 95% confidence intervals for R_t estimates.

I would encourage you to plot the raw data, i.e., the number of new cases over time for each outbreak. Furthermore, I couldn't find a definition of the start and end of an outbreak. How were they defined and were they defined consistently across outbreaks? Start and end date can influence the estimation of R_t and the estimated effects of public health measures if the dates are set too early or too late.

Response: We agree and have visualised the raw data in the following link: https://github.com/wxl1379457192/Zeroing_out_emerging_contagions/blob/main/Reported%20new%20cases%26Inferred%20infection.pdf, which includes the reported new cases, inferred infection and NPIs implementation over time for each of the 131 study outbreaks. Besides, the start and end date of the study outbreaks were defined

consistently in this study as below, and we have also clarified the definition in Section Methods (see lines 373-379).

“Based on the case information reported from official sources, we defined the onset of an outbreak as the date when new non-isolated cases started to increase. The end of an outbreak was defined as the date when zero new cases were initially reported, followed by a consecutive period of more than 7 days with no new infections. When an outbreak displayed two or more discernible peaks, with a sustained period of over 5 days between these peaks characterized by zero new cases, we designated the appearance of new cases as the beginning of a new wave.”

Steps 1-3) entail a lot of assumptions and priors, but the Supplements barely contain graphical displays of the corresponding distributions of priors. For example, the serial interval of each SARS-CoV-2 variant (Supplementary Table 3) could be plotted and compared in one figure. Similarly, I suggest adding displays for other delay distributions and important priors such as the effect of non-pharmaceutical interventions (NPIs). Graphical displays will make it easier for readers to assess the assumptions and follow the methods.

Response: Thanks for this comment. We have visualized all the prior distributions used in our Bayesian inference model, and updated the Supplementary Material then (see supplementary Fig 3-6).

Supplementary Fig 3. The prior distribution of the serial interval for each variant.

Supplementary Fig 4. The prior distribution of incubation period for each variant.

Supplementary Fig 5. The prior distribution of reporting delay (left) and the onset-to-reporting lag (right).

Supplementary Fig 6. The prior distribution of effect parameters of NPIs (left) and the correlation parameter of the control factors (right) in the Bayesian inference model.

The estimation of R_t is separated from the empirical estimation of the effects of public health measures, thereby ignoring uncertainty from step 1) in step 2). Separating step 1) and 2) is fine, but you should somehow incorporate uncertainty from step 1) in 2). Alternatively, you could perform 1) and 2) in a single model like others [2-3].

Response: Thank you for bringing this to our attention. Yes, it is important to transfer the uncertainty associated with the estimated R_t (step 1) into the Bayesian inference model (step 2), specifically, as the uncertainty in the distribution of R_t to be analyzed. To ensure a coherent analysis in this study, as suggested by the reviewer, we incorporated the mean and variance of the daily estimated R_t as inputs in step 2. We found that the improved model yielded an enhanced goodness-of-fit, with the R^2 value slightly increasing from 0.45 to 0.50. We have revised the corresponding part in Methods in line 429-445, which reads as “

$$\Phi_{t,v} = R_{0,v} \prod_{i=0}^n \exp(-\alpha_{i,v} X_{i,t,v} - \beta_i T_{i,t,v}) + \varepsilon$$

$$\Phi_{t,v} \sim \text{gamma}(R_{t,v}, \sigma_{t,v})$$

..... $\sigma_{t,v}$ indicates the variance of $R_{t,v}$, which was estimated by the observed cases data on a daily basis for each study outbreak”.

2) Estimating the effects of public health measures: I like that the authors collect a comprehensive dataset of public health measures that considers varying intensity of the measures. I am not sure though if it is good practice to normalize the ordinal NPIs into a variable ranging from 0 to 1. Most studies consider NPIs as binary indicator variables [4]. I think you could do the same by encoding each ordinal variable as multiple binary variables. It depends on how much variation you have between outbreaks whether you can estimate the effects of each level of an NPI. If not, you may use a less granular coding of NPIs.

Response: Agree, the quantification of NPIs is always a challenge to be resolved. Considering NPIs as binary indicator variables can help us to understand the effects of implementation of the corresponding NPIs. However, NPIs were generally deployed with a geographical scope, which was hardly be considered by such binary indicator variables. Under the situation, the implementation of NPIs with different geographical scopes would naturally yield different effects; whilst their effects were modeled consistently instead. In this study, the collected outbreaks were at the city level, but

sometimes NPIs were deployed at community level or district level. If we were to split the NPIs based on their ordinal levels, we would encounter situations where policies of specific geographical scope or intensity only existed during certain periods of variant circulation, making it difficult to compare the effects of the same policy across different time periods (see supplementary Figures file "NPI split plot.pdf" via https://github.com/wxl1379457192/Zeroing_out_emerging_contagions/blob/main/NPI%20split%20plot.pdf). Therefore, we transferred the implementations of NPIs into a variable ranging from 0 to 1 to reflect both the intensity of the NPIs (how strict the NPIs was informed) as well as their geographical scopes (the influenced population), revealing the virtual physical effects of the NPIs on population's behaviors. Such quantification of NPIs can longitudinally evaluate the effects of NPIs regarding the variants instead of each outbreak, given previous studies mainly focused on a relatively short period. For example, the 'Medicine Management' measures at level 2-3 were implemented after the emergence of the Delta variant.

Besides, to address these concerns and provide additional clarity, we have included further detailed information regarding the normalization process in the supplementary materials in SI line 76-88, as ***"In the post-Delta era, China is dedicated to implementing more precise zero-COVID policies to minimise societal impacts. Hence, geographic scope was taken into account in the normalization of measures, calculated as:***

$$x_j = (I_j - 1) + \frac{a_j}{\max(a_j)}$$

$$x'_j = \frac{x_j}{\max(x_j)}$$

Where x_j is the intervention with additional information on geographic scope, I_j indicates the intensity of each measure, a_j is the size of its geographic scope, and x'_j is the normalized measure. Other measures without additional information were normalized by min-max normalization, ranging from 0-1. For PCR-based mass screening, the results were weighted by the frequency of testing performed per week (= total number of tests/7 days). The normalized value, corresponding intensities and additional information including geographic scope and testing frequency for all ordinal variables can be found in Supplementary Table 2."

Additionally, we have added a table (see supplementary Table 2) that presents the normalized values and their corresponding policy intensities along with the additional information including geographical scope and testing frequency per week for all ordinal variables. This will facilitate a better understanding of the meaning and impact of each NPI at different values for readers. Nonetheless, we also put this issue as one of the limitations of this study to discuss, as ***"Considering NPIs as binary indicator variables have been commonly used, which can help us to quantify the stringency and effects of measures³⁻⁵. However, outbreaks were reported at the city or prefectural level, but interventions were often deployed at a smaller administrative unit such as community or district level in China. Moreover, when quantification was based on the ordinal levels of NPIs, we encountered situations where specific geographic areas or intensity policies only existed during certain periods of variant circulations. This makes it challenging to compare the effects of one policy across different time periods. For example, Medicine Management measures of mid-stringency (Require PCR tests for purchasers) were implemented after the emergence of the Delta variant. Therefore, to longitudinally evaluate the impact of NPIs on variants, we assembled the data of deployment geographical scope, thus combining them with***

the stringency of NPIs and transforming the space-varying interventions into a variable ranging from 0 to 1." in SI line 55-66.

Supplementary Table 2. Normalized NPI Lookup - Intensity and Geographic scope. Coding of additional information indicates geographic scope for business premises closures (BPC), public transportation closures (PTC), gathering restrictions (GR), workplace closures (WC), school closures (SC), facial masking (FM), and medicine management (MM), while indicates the total number of tests per week for mass screening (MS).

ID	Normalized intensity	Coding stringency	of Coding additional information	of ID	Normalized intensity	Coding stringency	of Coding additional information	of
L	0.00	0			0.00	0	0	
	0.20	1			0.13		1	
	0.40	2	NA		0.25	1	2	
	0.60	3			0.38		3	
	0.80	4		SC	0.50		4	
	1.00	5			0.63		1	
BPC	0.00	0	0		0.75	2	2	
	0.06		1		0.88		3	
	0.13		2		1.00		4	
	0.19	1	3		0.00	0		
	0.25		4	FM	0.50	1	NA	
	0.31		1		1.00	2		
	0.38	2	2		0.00	0	0	
	0.44		3		0.03		1	
	0.50		4		0.06		2	
	0.56		1		0.09		3	
PTC	0.63	3	2		0.11	1	4	
	0.69		3		0.14		5	
	0.75		4		0.17		6	
	0.81		1		0.20		7	
	0.88	4	2		0.23		1	
	0.94		3		0.26		2	
	1.00		4		0.29		3	
	0.00	0	0		0.31	2	4	
	0.06		1	MS	0.34		5	
	0.13	1	2		0.37		6	
MS	0.19		3		0.40		7	
	0.25		4		0.43		1	
	0.31		1		0.46		2	
	0.38	2	2		0.49		3	
	0.44		3		0.51	3	4	
	0.50		4		0.54		5	
	0.56		1		0.57		6	
	0.63	3	2		0.60		7	
	0.69		3		0.63		1	
	0.75		4		0.66	4	2	
0.81	4	1		0.69		3		

0.88		2	0.71		4
0.94		3	0.74		5
1.00		4	0.77		6
0.00	0	0	0.80		7
0.08		1	0.83		1
0.17		2	0.86		2
0.25	1	3	0.89		3
0.33		4	0.91	5	4
0.42		1	0.94		5
GR 0.50		2	0.97		6
0.58	2	3	1.00		7
0.67		4	0.00	0	0
0.75		1	0.08		1
0.83		2	0.17		2
0.92	3	3	0.25	1	3
1.00		4	0.33		4
0.00	0	0	0.42		1
0.13		1	MM 0.50		2
0.25		2	0.58	2	3
0.38	1	3	0.67		4
WC 0.50		4	0.75		1
0.63		1	0.83		2
0.75		2	0.92	3	3
0.88	2	3	1.00		4
1.00		4			

I don't think that references 13 to 16 constitute representative examples of previous studies on the effects of public health measures. For one thing, Ref 13 presents data rather than analysis and Ref 16 is a Matters Arising concerning the study of Flaxman et al. [2] and should probably cited together. I think that references 59 (Brauner et al.) and 61 (Haug et al.) would be more prominent examples of this stream of literature. You may further draw from the list of studies reviewed by Banholzer et al. [4] for possible additions to your related work.

Response: Thanks for the comment. We have added more representative studies into the reviewed introduction. Following the reviewer's suggestion, we have also drawn from the list of studies reviewed by Banholzer et al. [4] to identify additional sources that can enhance the comprehensiveness of our literature review. The updated references are listed below:

1. Flaxman, S. *et al.* Estimating the effects of non-pharmaceutical interventions on COVID-19 in Europe. *Nature* **584**, 257–261 (2020).
2. Brauner, J. M. *et al.* Inferring the effectiveness of government interventions against COVID-19. *Science (1979)* **371**, (2021).
3. Banholzer, N. *et al.* The methodologies to assess the effectiveness of non-pharmaceutical interventions during COVID-19: a systematic review. *Eur J Epidemiol* **37**, 1003–1024 (2022).
4. Lison, A. *et al.* Effectiveness assessment of non-pharmaceutical interventions: lessons learned from the COVID-19 pandemic. *Lancet Public Health* **8**, e311–e317 (2023).

5. Banholzer, N. *et al.* Estimating the effects of non-pharmaceutical interventions on the number of new infections with COVID-19 during the first epidemic wave. *PLoS One* **16**, e0252827 (2021).
6. Peak, C. M., Childs, L. M., Grad, Y. H. & Buckee, C. O. Comparing nonpharmaceutical interventions for containing emerging epidemics. *Proc Natl Acad Sci U S A* **114**, 4023–4028 (2017).
7. Haug, N. *et al.* Ranking the effectiveness of worldwide COVID-19 government interventions. *Nat Hum Behav* **4**, 1303–1312 (2020).

It is very interesting to compare the effects of public health measures between different SARS-CoV-2 variants. To my knowledge, this is quite unique and it is also important to understand how the effectiveness of measures depends on the variant, or more generally, the time and context in which they were implemented. I have only a minor suggestion here. Instead of splitting the data into groups by variant, the authors could analyze them jointly with a hierarchical model using varying slopes on the effects of public health measures for each variant. Thereby, information across outbreaks for the estimation of all non-variant specific parameters are shared, which should be more efficient for model estimation.

Response: Thanks for this comment. Based on the suggestion, we have revised our workflow of the Bayesian inference model, where the data of different variants were processed jointly. Specifically, we implemented a hierarchical model with varying slopes to account for the effects of NPIs for each variant. This approach allows information to be shared across outbreaks and variants, to estimate the effect of control factors more efficiently. We found that this modification enhances the robustness and rationality of assessing the impact of environmental factors. We have revised the manuscript accordingly in line 423-445, as

*“Specifically, a generalized linear relationship between NPI intensity and the reduction was considered below. The relationship between each individual NPI and effect on the reduction of R_t was assumed to be linear. This assumption was incorporated into a hierarchical model, where varying slopes were used for each variant. Non-informative priors with a Gamma distribution were included in the model consistent with methodology employed by Yong *et al.* and others^{20,24,54,69}.*

$$\Phi_{t,v} = R_{0,v} \prod_{i=0}^n \exp(-\alpha_{i,v} X_{i,t,v} - \beta_i T_{i,t,v}) + \varepsilon$$

$$\Phi_{t,v} \sim \text{gamma}(R_{t,v}, \sigma_{t,v})$$

Where $X_{i,t,v}$ indicates each individual NPI i under variant v on day t . n represents the number of NPIs. $T_{i,t,v}$ indicates the control variable, i.e., air temperature, population density and practical vaccination rate used in this study. $\alpha_{i,v}$ represents the coefficient of the covariate X across different variants v . Following the assumption by Flaxman *et al.*¹⁴, the NPI coefficients are assigned a Gamma distribution with shape parameter 1/6 and scale parameter 1 and further shifted by $\log(1.05)/6$ to accommodate for both positive and negative effects, see supplementary Fig 5. β_i represents the coefficient of the control variable. We placed it a Normal prior with a coefficient of $N(0, 0.5)$, indicating β_i might also be associated with the trajectory but remains constant across all variants. The contribution of other unobserved confounding factors to the R_t decay is indicated by the residual ε . We set the prior of the basic reproduction number $R_{0,v}$ to obey a Gamma distribution with a shape parameter of f and a rate parameter of 0.1, where f varied with SARS-CoV-2 lineages (see supplementary Table 5). $\sigma_{t,v}$ indicates the variance of $R_{t,v}$, which was estimated by the observed cases data on a daily basis for

each study outbreak.”

The reporting of the effects of public health measures with median and IQR is rather unusual. I think the vast majority of studies report mean/median and 95% credible interval. I can see that a leave-one-out cross validation was performed in the Supplements, thus you could also report prediction intervals as in Brauner et al. [3], which reflects even greater uncertainty in the estimates.

Response: Thank you for pointing out this issue. We have updated our results with mean and 95% credible intervals. The estimations of NPIs effect, which was validated by leave-one-out cross-validation, were reported by prediction intervals now, as in Brauner et al. [3].

The formula on page 19 line 388 probably has one “exp” too much.

Response: Thank you for bringing this to our attention. We have removed the redundant “exp” from the formula.

In Supplementary Text A.2, I did not quite understand what exactly a_j is (the size of the geographic scope). Is it the size of the population? If you want to consider that only a subpopulation was affected by the measure, then you could weigh the effects of public health measure by the proportion of the population that is affected, as e.g. in Banholzer et al. [5].

Response: Due to the gradual relaxation of non-pharmaceutical interventions (NPIs) in different geographic areas in China, we introduced a specific indicator to describe the concept of “precise control” frequently mentioned by the Chinese government. This geographic scope indicator represents the primary regional extent of policy implementation.

In each Chinese provincial region (administrative unit 1), geographically adjacent communities/villages normally form a street or township (admin unit 4), and several neighboring streets/townships constitute a district or county (admin unit 3). Cities/prefectures (admin unit 2) usually consist of several urban districts and surrounding counties. Taking the business premises closure (BPC) as an example, the geographic scope Level 1 indicates that only the shops/business within the affected community need to be closed or strictly controlled for entry and exit. When the impact of the outbreak expands, shops within the streets composed of relevant communities must implement corresponding control measures, representing Level 2. When the policy extends to the district or county level, the geographic scope is classified as Level 3. If the same level of control measures is uniformly implemented across the entire city/prefecture, the level is upgraded to Level 4.

The geographic range serves as an indicator to describe the size of the affected group. However, we did not utilize population size as an indicator for two reasons. First, it is challenging to obtain population proportions at more precise geographic scales such as communities or streets. The data collection process would require specific boundary shapefiles and the number of residents at a community, which is not publicly available. Second, public health measures for COVID-19 are often not formulated based on population proportions. In China, policy implementation generally takes administrative divisions as the basic unit.

However, to further clarify this issue, we have added descriptions in the supplementary material (see lines 26-37), which can help readers understand the meaning represented by the geographic scope indicator for each level, as

“As the implementation of COVID-19 intervention policy might vary in stringency across geographic areas/administrative divisions within a city (prefecture level), we also used another ordinal metric to denote the geographic range (1 - Community/village level, 2 - Township level, 3 - District/county level, 4 - City/prefecture level) for BPC, PTC, GR, WC, SC and MM measures, where the most stringent NPI was presented. During the pandemic, cities in China typically manage communities or villages as the basic units. Several geographically adjacent communities/villages form a street or township, and several neighboring streets/townships constitute a district or county. Cities/prefectures usually encompass multiple urban districts and surrounding counties. Therefore, a higher value of the geographic scope indicator means that a larger area or population would be affected by the measures implemented.”

The prior for the effects of social distancing measures could be borrowed from Brauner et al. [3]. They use an asymmetric prior where positive effects (reduction in Rt) are more probable, but the prior still allows for negative effects (increase in Rt).

Response: Thanks for this suggestion. In the revised manuscript, we have used the prior from Flaxman et al.[2]. Following the same assumption, we have modified the prior for the NPI effects parameters to allow for a certain degree of negative effects while still favoring positive effects, in the manuscript now read as *“Following the assumption by Flaxman et al., the NPI coefficients are assigned a Gamma distribution with shape parameter 1/6 and scale parameter 1 and further shifted by $\log(1.05)/6$ to accommodate for both positive and negative effects”*. The updated prior is as follows:

$$\alpha_{i,v} \sim \text{Gamma}(1/6, 1) - \frac{\log(1.05)}{6}$$

After comparing the results with the original prior, we have found that modifying the prior information had little effect on the final outputs. Nevertheless, we agree that incorporating this modification may improve the validity of our model.

I think the third and fourth limitation of the study should be placed more prominently before or in the Result section. It is important context regarding the magnitude of the effects that they are conditional on the already implemented long-lasting measures. It is also important context that your exclusion criteria for outbreaks most likely underestimates the effectiveness of contact tracing or other measures that managed to control the outbreak quickly.

Response: We fully acknowledge that the exclusion of small outbreaks with rapid containment may lead to some underestimation of the effectiveness of contact tracing or other measures. However, we chose to exclude these data from our study for two main reasons. First, we found that estimating the Rt of the outbreak with a short duration resulted in poor robustness. Second, in the real world, non-pharmaceutical interventions may not be formulated in a rapid manner and implemented on a large scale in the early stages of outbreaks, caused by entirely new infectious diseases, when only few suspected cases are found. Following your suggestion, we have moved the third and fourth limitations into the Results section and have reorganized it as *“However, there are two key aspects that need to be considered when interpreting the above findings. First, this study did not assess the effects of long-lasting international travel restrictions and quarantine for reducing the introduction risk, which might overestimate the impact of other interventions in containing local transmission within each city. Second, the small-scale, short-duration outbreaks were excluded from the modelling, which might have led to an underestimation of the effectiveness of some NPIs, such as contact tracing, that might also play an important role in controlling epidemics”*

in the early phases.” in line 174-181.

3) Scenario modeling: I cannot comment much on this step as it is not directly my area of expertise. I was a bit puzzled by the statement in line 172 that the results from the ISEIRV model validate the results from the Bayesian inference model. Anything else would surprise me given that the parameters for the effects of public health measures of the ISEIRV model are informed by the Bayesian inference model.

Response: Much of the previous studies used Bayesian inference models to evaluate the effects of NPIs by attributing the variation of R_t onto the variation of the NPIs implementation. Under such a circumstance, the effects of NPIs were super mixed into a single impact on the change of R_t , where the mechanisms of different NPIs were neglected. For example, facial mask wearing and gathering restrictions contained the transmission through the reduction of virus load per contact and the size of contact per capita, respectively. And therefore, we cannot reasonably conduct scenario modeling under combinations of epidemiological settings against different NPIs implementation contexts. With a compartmental model, i.e., ISEIRV in this study, we can further examine the effects of different NPIs in a more detailed fashion, and produce more practical conclusions for the policy-makers. Given we considered NPIs for different parameters of the ISEIRV, the relationship between all the NPIs and R_t was not guaranteed to be consistent to the one in our Bayesian inference model (all the study NPIs mutually independent to each other). Thus, the validation of the results from the Bayesian inference model by the results from the ISEIRV can confirm the reliability of introducing the effects of NPIs estimated by the Bayesian inference model into their impact on different parts of the dynamics of Covid-19 transmission.

I commend the authors for the time and effort invested in this study and wish them best of luck with any revisions. I hope my comments are helpful for this.

Response: Many thanks for your time and effort in reviewing our work. We have tried our best to adequately address your valuable comments and suggestions, and we believe that the quality of our manuscript has been significantly improved as a result.

Reviewer #2 (Remarks to the Author):

General comments:

1. Many thanks to the authors for an interesting set of analyses.

Response: We appreciate the reviewer's very positive and thoughtful comments.

2. I suggest adding some more specificity in the descriptions of the NPIs in the main paper, particularly the mass screening and contact tracing scenarios. The supplemental information provides much more detail (particularly in lines 30-44 in the supplement). While the terms “mass screening” and “contact tracing” are familiar in general, I think it would be helpful for readers to see the specific descriptions associated with China's implementation(s) of the zero-COVID policy (e.g., the implementation of contact tracing isn't defined until lines 250-251. This is particularly relevant for mass screening which was implemented in very different ways at different times in different countries.

Response: To address this concern, we have added more descriptions of the scenarios of MS and CT in lines 389-397, now read as *“In China, MS and CT were defined differently as essential approaches for infection detection and tracing. Mass screening was*

typically conducted for individuals without a clear exposure history but who were at risk of exposure, such as those residing in the same community as a confirmed case. Only if a positive test result was obtained, the individual would be quarantined, and until then they were allowed to move around. However, close contact tracing refers to the isolation of individuals with a history of exposure to a case. These individuals were regularly tested for COVID-19 nucleic acid during the quarantine period.”

3. In addition to the definitions requested in the previous general comment, I also suggest adding some brief discussion of the role of mass screening and contact tracing NPIs compared to social distancing and masking. Social distancing and masking are general requirements applied to all individuals in an area, regardless of their own infection or exposure status while contact tracing and mass screening focus on isolating detected cases or exposed individuals respectively. This difference between NPIs applied to all or NPIs designed to detect individuals who will then be subject to isolation seems important in the discussion of effectiveness early or late in an emerging pandemic. If there are many scattered cases, we might expect SD and M to work better than MS and CT, which is what was observed.

Response: Thanks reviewer for the comments and insights. We have added a separate paragraph to discuss the generalizability of the findings of different measures as well as relevant findings in other countries. Particularly, the feasibility and challenges associated with those NPIs. See lines 298-318:

“...For instance, similar to findings in the United Kingdom³⁶ and the United States³⁷, contact tracing was found to be more effective than social distancing and large-scale tests at the early stages of outbreaks, in curbing transmission by pinpointing the infected individuals and isolating close contacts in communities³⁸⁻⁴¹, while social distancing was general requirements applied to all individuals in an area, regardless of people’s infection or exposure status. However, if there are many scattered cases or a significant proportion of the population infected in the middle to late stages of the outbreak, we might expect social distancing and masking to work better than mass screening and contact tracing^{36,42,43}. Moreover, in places with considerable population flow, such as airports, contact tracing and mass screening might be more effective than social distancing and masking in reducing imported infections^{39,44}. In addition, mass testing could be increasingly laborious and time-consuming facing a large population size. Pre-symptomatic or asymptomatic infections might also weaken the impact of population-scale testing, as many infectious individuals cannot be identified or the gatherings in test sites could also increase the transmission⁴⁵. Nevertheless, facial masking played a most stable role, especially given its role in reducing the risk of transmission in households or indoor settings⁴⁵⁻⁴⁹. In light of this, it is evident why governments worldwide have responded to the emergence of COVID-19 by enacting legislation mandating mask usage in public places⁵⁰. Additionally, the continued significance of mask-wearing as a vital tool in East Asia to control the spread of respiratory infections is well-documented⁵¹”

4. Specific comments:

-Lines 43-45 (and throughout). The effects are described as percentages without specifying that these are percent reductions in the instantaneous reproductive number. While it may seem repetitive, I suggest clarifying that the effect is “a 23% reduction in the instantaneous reproductive number” on line 43, and “xx% reduction” in the following sentences. Currently the effects are described without context or interpretation.

Response: Thanks for your reminder. We have made the necessary corrections to provide more context and interpretation regarding the effects of NPIs.

-Line 45. Omit “However”.

Response: Done.

-Line 58. “so as to avoid developing”. Should this be “so as to avoid the epidemic developing”?

Response: Yes, it has been corrected.

-Line 71. I suggest replacing “such as” to “including”. Mental health is often overlooked among lists of secondary health effects so I suggest “including” it rather than listing it as an example.

Response: Agree. We have made the suggested change.

-Line 128. I suggest referencing supplemental figure 1 (the map) here as well.

Response: Done.

-Line 129. “intensity of lockdown”. I suggest adding more detailed description of the (0,1) assessment of intensity in the main text prior to its use here. The parenthetical definition in line 131-133 is fairly vague and details in the supplemental information are not referenced here. This could be accomplished with a brief (2-3 sentence) paragraph setting the stage.

Response: We appreciate the suggestion. In the reviewed manuscript, we reorganized the relevant description of intensity of lockdown, and provided more detailed for better understanding in line 129-135, as *“For example, on a regional average, the intensity of lockdown in South Central and Southwest China decreased from 0.5 in Pre-Delta era to 0.4 in Omicron era, while mass screening had an intensity increasing from 0.1 to about 0.3 in most regions (Note that the intensity of each measure was normalised from 0 to 1, where 1 indicates the strictest and 0 indicates the least strict). This indicates that affected cities in southern China eased some stringent measures, such as lockdown, and increased the frequency of PCR testing.”*

Furthermore, we have supplemented a Lookup table in supplementary that includes information on the normalized values, their corresponding intensity and geographical scope for each NPI (Please also see supplementary Table 2 in the response to Reviewer 1’s comment #2).

-Line 206. I suggest omitting “relatively”.

Response: Done.

-Line 273. What does “facial masking is set to 0.5” mean in terms of implementation? It is the middle of the (0,1) range but since masks were mandatory, I’m not sure if this refers to compliance? I suspect it does not mean only 50% of individuals were wearing masks, but it would be helpful to calibrate this mid-level assessment in light of the entire zero-COVID policy.

Response: Thank you for raising this question. We would like to emphasize that the

quantities of NPIs used in our study is related to both the intensity of individual NPIs and the geographic scale as they were implemented at. Therefore, the quantities were considered to capture the virtual impact of such NPIs implementation i.e., how many populations would really change their behaviors to what extent as a result of particular NPIs deployment. For facial masking, it is generally deployed with intensities 0 (not mentioned, rare in zero-COVID policy), 1 (recommended to wear), and 2 (required to wear). After normalization of NPIs intensities, it became 0 (not mentioned, rare in zero-COVID policy), 0.5 (recommended to wear), and 1 (required to wear). Noting that there is no geographic scale for facial masking, and the number of facial masking refers to compliance here. Given it is scenario modeling, considering the realizability and the generalizability to the other countries/areas, we thus intentionally set a mid-level assumption, i.e., 50% probability that an individual wearing a mask amid contact with an infected person).

The related sentence has been revised as ***“We assumed a 50% probability that an individual wearing a mask amid contact with an infected person, considering the realizability and the generalizability to the other countries/areas”*** in line 282-284. Furthermore, we have incorporated additional scenarios assuming a higher (75%) and lower (25%) probability that a wearing-mask individual comes into contact with an infected person in SI (see Supplementary Figure 19-36).

-Line 287. “The prompt” to “Prompt”.

Response: Done.

-Line 308. “The data of” to “The data from”.

Response: Done.

- Line 344. “Data sources of” to “Data sources for”.

Response: Done.

- Line 355. “the local Centers for Disease Control and Prevention”. Does this refer to Province-level offices? This is not clear to me.

Response: Thanks for this question. In China, almost each administrative area at national, province, city/prefecture, and district/county levels have a Center for Disease Control and Prevention (CDC), responsible for disease surveillance, prevention, control, and response activities within its area. Local CDCs refer to the province, city/prefecture, and district/county-level CDCs, which are under the guidance and supervision of the national CDC (China CDC). To clarify this, we added more details in line 366-368, as ***“Data sources for outbreaks included press releases from local government websites or reports from the local disease control and prevention agencies at the province, city/prefecture, and district/county levels...”***

- Line 386 and following. The relationships have linear components but most are transformed so I suggest referring to “generalized linear” relationships throughout.

Response: We have revised the manuscript according to your recommendation, and we now refer to the relationships as “generalized linear” throughout the relevant sections.

-Line 395. “the coefficients are placed as a Gamma prior with hyperprior, following $\gamma(u,1)$ and $u \sim \text{uniform}(0,1)$.” I suggest “the coefficients are assigned $\gamma(u,1)$ ”

prior distributions and u assigned a uniform(0,1) hyperprior distribution". Also, a gamma prior on association parameters would only allow positive estimates of association parameters, is this what is intended?

Response: Following your and the first reviewer's suggestion, we have taken it into consideration to borrow the prior for the effects of NPIs from Flaxman et al [2]. With the same assumption, we have modified the prior for the NPI effects parameters to allow for a certain degree of negative effects while still favoring positive effects, as we reorganized in the manuscript "*Following the assumption by Flaxman et al.14, the NPI coefficients are assigned a Gamma distribution with shape parameter 1/6 and scale parameter 1 and further shifted by $\log(1.05)/6$ to accommodate for both positive and negative effects,*". The updated prior is as follows:

$$\alpha_{i,v} \sim \text{Gamma}(1/6, 1) - \frac{\log(1.05)}{6}$$

After comparing the results with the original prior, we have found that modifying the prior information had little effect on the final outcomes. Nevertheless, we agree that incorporating this modification improves the scientific validity of our model.

- Line 410. Is the underlining of $X_{i,t}$ intentional? Or is there supposed to be a bar on $X_{i,t}$ in both places to indicate the average?

Response: Sorry for the confusion. It was a typo, and we have corrected it to $\overline{X_{i,t,v}}$ now.

- Line 412. "normalized NPI". I assume this refers to the (0,1) scoring of the NPIs but the reference to normalization is not clear until deep into the supplemental information. If a brief description and interpretation of the (0,1) range of values for NPI is added earlier on (as suggested in a few comments above), there would not be a need for the "normalized" qualifier.

Response: Thank you for the suggestion. We have removed the "normalized" qualifier and added a lookup table in the supplemental information that provides a brief description and interpretation of the (0,1) range of values for NPI. Please see *supplementary Table 2*. This table will help readers understand the corresponding NPI intensity and the geographical scope for each value.

- Line 419. "Divided" to "divides".

Response: Corrected.

- Line 446. "ratio of facial masking". As in other comments, clearly defining the NPI scores used would be helping. Here "ratio of facial masking" does not refer to the fraction masking. Also, how is "efficiency of facial masking for preventing indoor transmission" defined and implemented in the model? I don't believe the authors' model differentiates between indoor and outdoor transmission, does it?

Response: Sorry for the confusion. We have defined all the study NPIs in *supplementary Table 2*. We used the 'ratio' of facial masking to assume the proportion of the population that would wear masks, i.e., compliance as you pointed before. Specifically, 0 (not mentioned, rare in zero-COVID policy), 0.5 (recommended to wear, assuming 50% population wearing masks), and 1 (required to wear, assuming the whole population wearing masks). Given there is little evidence of the empirical effects of

wearing masks in China, we used the efficiency of facial masking for preventing indoor transmission where the measure is most effective, as documented in the literature [1], to represent the general effect of mitigating transmission by wearing masks. The 'ratio' of facial masking, as the intensity of implementing this measure, was used in the Bayesian inference model to evaluate its relative effects during the Zero-covid policy. Then, in the ISEIRV model, the 'ratio' of facial masking was further multiplied by the *efficiency of facial masking for preventing indoor transmission* in order to capture the proportion of the Susceptible that wouldn't be exposed to the virus in simulations. The words have been revised as "the fraction of the population that complied with the requirement to wear a mask" now.

[1] Coclite, D. et al. Face Mask Use in the Community for Reducing the Spread of COVID-19: A Systematic Review. *Front Med (Lausanne)* 7, (2021).

- Line 451. "were linearly" to "was linearly".

Response: Corrected.

- Line 454. "the weights...was" to "the weights...were". Also, the phrase "the weights were determined by their effects". What do the authors mean by "effects"? It would seem circular to weight based on the estimated effects but I feel I'm missing something.

Response: The weight here is used to calculate the intensity of the integrated measure of contact reduction based on the intensities of several relevant NPIs. The weights can be considered as a prism reflecting the relative size of the effect of individual NPIs on Covid mitigation; thus, if some NPIs contributed larger reductions in R_t , their weights would naturally be larger also compared to the weights of other NPIs that contributed smaller reductions. In this way, the intensity of the general measure of contact reduction would be determined by the intensities of those NPIs with a larger impact on Covid mitigation and better reflect the effect of integrated measure of contact reductions. Therefore, we argue that the weights were determined by their effects. To make it clearer, the related content has been revised as:

"The intensity of the general contact-reduction measure was then defined by the linear combination of the intensities of relevant NPIs on a daily basis:

$$x_1 = \sum_{i=1}^6 w_i * X_i$$

$$w_i = \frac{e_i}{\sum_{i=1}^6 e_i}$$

where the weights w_i for each NPI X_i was proportionally determined by its empirical e_i of reducing R_t assessed by the Bayesian inference model."

- Line 465. "recovery rate then" to "recovery rate".

Response: Done.

- Line 498. What does "GEE" refer to here?

Response: GEE refers to Google Earth Engine. Thank you for bringing me to this abbreviation. We have added the full name of it in line 542 now.

- Supplement, line 56. What do the parantheses denote here?

Response: There should be a "max()" function in the formula. The formula has been corrected accordingly. Thank you for bringing this to our attention.

- Supplement. The Bayesian inference model description is presented across a long narrative, I would find it helpful to have equations for all of the elements (equations, priors, hyperpriors) in a Box to present along with the narrative so all of the elements are in one place and the model definition is separated from details of its implementation. I feel this would provide a good context for the listing of the sensitivity analysis priors on lines Supplement 413-418.

Response: We appreciate your comment. We have added a *Supplementary Equation Table*(https://github.com/wxl1379457192/Zeroing_out_emerging_contagions/blob/main/Supplementary_equation_table.doc) to present all the equations, priors, hyperpriors, and details of the Bayesian inference model. By doing so, we aim to enhance the readability and facilitate a better understanding of the model and its sensitivity analysis.

Supplementary Equation Table. Equation summary of the Bayesian inference model, including formulas, the corresponding elements, and the priori information.

	Equation	Elements	Prior distribution
Rt estimation	$E(I_t) = R_t \sum_{k=1}^t I_{t-k} W_k$	I_t—the number of infections at time t, which is inferred by the reported cases by removing the time lag t' from exposure to reporting. This time lag t' is the sum of the incubation period (t_c') and the reporting delay (t_r'). W_k—the infectivity profile which depends on the serial interval R_t—the instantaneous reproduction number at time t I_{t-k}—is the incidence at time t-k	Serial interval: Original: Gamma(5.80, 3.20) Alpha: Gamma(5.80, 3.20) Delta: Gamma(5.80, 3.20) Omicron: Gamma(5.80, 3.20) Incubation period (t_c'): Original: Lognormal(1.78,0.52) Alpha: Lognormal(1.50,0.46) Delta: Lognormal(1.25,0.34) Omicron: Lognormal(1.02,0.45) Report delay (t_r'): Before mass screening: Lognormal(0.82,0.84) After mass screening: Binomial(1,0.3)
NPIs effect estimation	$\Phi_{t,v} = R_{0,v} \prod_{i=0}^n \exp(-\alpha_{i,v} X_{i,t,v} - \beta_i T_{i,t,v}) + \varepsilon$ $\Phi_{t,v} \sim \text{gamma}(R_{t,v}, \sigma_{t,v})$	$R_{0,v}$—the basic reproduction number of variant v $X_{i,t,v}$—each individual NPI i under variant v on day t $T_{i,t,v}$—the control variable, i.e. air temperature, population density and practical vaccination rate $\alpha_{i,v}$—the coefficient of the covariate X across different variants v β_i—the coefficient of the control variable $R_{t,v}$—the instantaneous reproduction number at time t of variant v $\sigma_{t,v}$—the variance of $R_{t,v}$	$R_{0,v}$: Original: gamma(3.32,0.1) Alpha: gamma(4.28,0.1) Delta: gamma(4.90,0.1) Omicron: gamma(9.05,0.1) $\alpha_{i,v} \sim \text{gamma}(1/6,1) - \log(1.05/6)$ $\beta_i \sim \text{normal}(0,0.5)$

- Supplement, lines 299-307. Is it fair to refer to this approach as approximate Bayes' computation (ABC)? Parameters generated from priors, model generates output that updates the prior to yield the posterior?

Response: Yes, the work flow of seeking the parameters was in a Bayesian fashion. We have further clarified this point in the supplement Section C.1 Calibration of ISEIRV model, see lines 351-352. It reads as ***"A Bayesian optimisation framework based on the approximate Bayes' computation was developed to optimize quantization hyperparameters in ISEIRV models"***

- Supplement, line 311. "splitted" to "split".

Response: Corrected.

- Supplement, line 329. The adjusted R-squares above 99% seem remarkably high. Is this common in model validation?

Response: We appreciate your observation and concern regarding the high adjusted R-

squares above 99% in our model validation. We apologize for the oversight in Figure S13a, where the case number shown represents the cumulative cases for each outbreak, including the training set and the test set, rather than only the test set. This discrepancy has led to the inflated R-squared values. To address this issue, we have corrected Figure S13a to display the cumulative cases specifically for the test set and recalculated the R-squared values (0.47) in the test set. Meanwhile, we have also validated the trend accuracy for simulating each outbreak and R² values range from 0.23 to 0.98, as shown in supplementary S13b.

Supplementary Fig 13. The performance of the ISEIRV model in simulating cumulative infections (a) and daily infections (b). The scatterplot (a) compared the simulated cumulative infections versus the observed cumulative infections for 131 outbreaks in each group for each variant. The boxplot (b) summarized the NRMSE and R² of the simulated daily infections found from COVID-19 propagation modelling during the Per-Delta, Delta and Omicron periods. The scatter in each box plot represented the point-to-point simulated precision of each outbreak.

-Supplement, lines 392 and following. The strategy comparison visualizations are quite interesting, thank you for sharing (they may well merit a publication on their own).

Response: Thanks for your positive feedback on the strategy comparison visualizations presented in the supplement. These visualizations were provided as additional figures to enhance the clarity and aesthetics of Figure 4 in the manuscript. While they were not included in the main text, we are grateful for your recognition of their potential value.

Reviewer #3 (Remarks to the Author):

This is an interesting study that gathered detailed COVID-19 and NPI data for 131 outbreaks occurred from April 2020 to May 2022 in China. While many previous works examined the effects of NPIs on COVID-19 in different countries and settings, this study is unique given the zero-COVID policies strictly enforced during the study period in China. The study first estimated the effects of ten interventions using a Bayesian inference framework and then used a compartmental model to simulate the effects of NPI combinations under different scenarios. Findings were effectively communicated using customized visualization tools (e.g., Fig. 4). Overall, I think this is a significant contribution to the literature on quantifying NPI effects on infectious disease outbreaks, which provides new insights in the context of eliminating emerging and re-emerging pathogens. I have a few comments that hopefully can help further improve the manuscript.

1. The study used an intensity level for interventions defined based on a set of criteria. For a better context, it would be useful to have some examples in the main text to show the full spectrum of intervention intensities. For instance, what does an intensity of 0.3 mean?

Response: Thank you for the comment. We have incorporated a table (*supplementary Table 2*) in the manuscript. This table presents the normalized values of interventions, along with their corresponding policy intensities. It also includes additional information such as the geographical scope and testing frequency per week for all ordinal variables. This can provide a comprehensive understanding of the meaning and impact of each NPI at different intensity values for readers. We believe that this table will enhance the context and facilitate a better interpretation of intervention intensities in our study.

Supplementary Table 2. Normalized NPI Lookup - Intensity and Geographic scope. Coding of additional information indicates geographic scope for business premises closures (BPC), public transportation closures (PTC), gathering restrictions (GR), workplace closures (WC), school closures (SC), facial masking (FM), and medicine management (MM), while indicates the total number of tests per week for mass screening (MS).

ID	Normalized intensity	Coding of stringency	Coding of additional information	ID	Normalized intensity	Coding of stringency	Coding of additional information
L	0.00	0	NA	SC	0.00	0	0
	0.20	1			0.13	1	
	0.40	2			0.25	1	2
	0.60	3			0.38	1	3

	0.80	4			0.50	4
	1.00	5			0.63	1
	0.00	0	0		0.75	2
	0.06		1		0.88	3
	0.13		2		1.00	4
	0.19	1	3		0.00	0
	0.25		4	FM	0.50	1
	0.31		1		1.00	2
	0.38		2		0.00	0
	0.44	2	3		0.03	1
BPC	0.50		4		0.06	2
	0.56		1		0.09	3
	0.63		2		0.11	4
	0.69	3	3		0.14	5
	0.75		4		0.17	6
	0.81		1		0.20	7
	0.88		2		0.23	1
	0.94	4	3		0.26	2
	1.00		4		0.29	3
	0.00	0	0	MS	0.31	4
	0.06		1		0.34	5
	0.13		2		0.37	6
	0.19	1	3		0.40	7
	0.25		4		0.43	1
PTC	0.31		1		0.46	2
	0.38		2		0.49	3
	0.44	2	3		0.51	4
	0.50		4		0.54	5
	0.56		1		0.57	6
	0.63	3	2		0.60	7

	0.69		3		0.63		1
	0.75		4		0.66		2
	0.81		1		0.69		3
	0.88		2		0.71	4	4
	0.94	4	3		0.74		5
	1.00		4		0.77		6
	0.00	0	0		0.80		7
	0.08		1		0.83		1
	0.17		2		0.86		2
	0.25	1	3		0.89		3
	0.33		4		0.91	5	4
	0.42		1		0.94		5
GR	0.50		2		0.97		6
	0.58	2	3		1.00		7
	0.67		4		0.00	0	0
	0.75		1		0.08		1
	0.83		2		0.17		2
	0.92	3	3		0.25	1	3
	1.00		4		0.33		4
	0.00	0	0		0.42		1
	0.13		1	MM	0.50		2
	0.25		2		0.58	2	3
	0.38	1	3		0.67		4
WC	0.50		4		0.75		1
	0.63		1		0.83		2
	0.75		2		0.92	3	3
	0.88	2	3		1.00		4
	1.00		4				

2. Line 176. The Bayesian inference revealed some synergistic effects between interventions. It seems the effects of NPIs were assumed independent and multiplicative

on R_0 (see the equation in line 388). Could the authors elaborate more on how synergistic effects were quantified?

Response: Sorry for any confusions caused by the expression used in the manuscript. In our study, when referring to synergistic effects in Figure 1, we were actually referring to the joint effects of multiple NPIs. To avoid misunderstanding, we have updated all the “synergistic” into “joint”.

Actually, we agree that there might be certain synergistic effects among NPIs. However, for the purpose of our analysis, we categorized NPIs based on their mechanisms of action and made the assumption of independence among the effects of different NPIs. We recognized the limitation of this assumption in the Discussion section, as *“Third, our conclusions resulted from the assumption of independent effects of NPIs. However, it is important to recognize that public health measures with different mechanisms may exhibit synergistic effects, such as mask wearing and social distancing, or vaccination with NPIs.”*

3. Small-scale outbreaks are highly stochastic, especially for emerging contagions at the early phase. I am wondering if simulations using the ISEIRV model considered such dynamical stochasticity and if this will impact the results. Maybe the authors can integrate the compartmental model stochastically using some distributions for new infections.

Response: Thank you for this comment. Our simulations considered various epidemiological parameters, including different incubation periods, infectious periods, R_0 values, and intervention scenarios, which allow our findings to be more broadly applicable to other emerging respiratory infectious diseases beyond COVID-19. However, we did not use the ISEIRV model to stochastically simulate very small-scale outbreaks (<50 cases) under the real-world containment scenarios for three reasons. First, to keep consistent with the study outbreaks, we only include the data of outbreaks as the Bayesian inference model. Second, small outbreaks with a short duration may easily die out in simulations, resulting in limited data samples and a reduced robustness of the results. Third, in reality, non-pharmaceutical interventions may not be formulated in a rapid manner and implemented on a large scale in the early stages of outbreaks, caused by entirely new infectious diseases, when only few suspected cases are found. Nevertheless, our simulations under counterfactual scenarios did yield some small-scale outbreaks, which can provide insights/evidence for tailoring response strategies under different scenarios of transmissibility, interventions, and population sizes.

4. It would be good to discuss the generalizability of the findings in other countries. Particularly, the feasibility and challenges associated with those NPIs.

Response: Thank you for pointing out this. According to this comment, in our revised manuscript (L.294-318), *“The relative effect of public health measures on zeroing out transmission was sensitive to the infectivity of pathogens, the timing and intensity of NPIs, and combinations of interventions, which should be considered in tailoring response strategies at different stages of an emerging epidemic or pandemic in various regions^{14,15,35}. For instance, similar to findings in the United Kingdom³⁶ and the United States³⁷, contact tracing was found to be more effective than social distancing and large-scale tests at the early stages of outbreaks, in curbing transmission by pinpointing the infected individuals and isolating close contacts in communities³⁸⁻⁴¹, while social distancing was general requirements applied to all individuals in an area, regardless of people’s infection or exposure status. However, if there are many scattered cases or a significant proportion of the population infected in the middle to late stages of the outbreak, we*

might expect social distancing and masking to work better than mass screening and contact tracing^{36,42,43}. Moreover, in places with considerable population flow, such as airports, contact tracing and mass screening might be more effective than social distancing and masking in reducing imported infections^{39,44}. In addition, mass testing could be increasingly laborious and time-consuming facing a large population size. Pre-symptomatic or asymptomatic infections might also weaken the impact of population-scale testing, as many infectious individuals cannot be identified or the gatherings in test sites could also increase the transmission⁴⁵. Nevertheless, facial masking played a most stable role, especially given its role in reducing the risk of transmission in households or indoor settings⁴⁵⁻⁴⁹. In light of this, it is evident why governments worldwide have responded to the emergence of COVID-19 by enacting legislation mandating mask usage in public places⁵⁰. Additionally, the continued significance of mask-wearing as a vital tool in East Asia to control the spread of respiratory infections is well-documented⁵¹”.

We highlighted that the effectiveness of NPIs might vary depending on the epidemiological and socio-economic contexts in other regions. Specifically, we emphasized the importance of considering the difference between NPIs applied to all individuals and those designed to detect and isolate specific cases or exposed individuals. Regarding the feasibility and challenges associated with NPIs, we also discussed the observed effectiveness of social distancing and masking (SD and FM) in China and other regions when there were only scattered cases. These measures are expected to work better in such scenarios. However, in areas with high human mobility, contact tracing (CT) and entry screening may be more effective than SD and FM in reducing imported infections. We acknowledged the challenges of implementing mass testing, such as the laborious and time-consuming nature, and the potential limitations when dealing with pre-symptomatic or asymptomatic infections. We also discussed the potential adverse consequences of long-term social distancing policies, including negative socioeconomic and mental health impacts.

5. A typo in line 388, an extra “exp” in the equation.

Response: Corrected.

6. Here is a relevant study that may be useful: Peak, C.M., Childs, L.M., Grad, Y.H. and Buckee, C.O., 2017. Comparing nonpharmaceutical interventions for containing emerging epidemics. PNAS, 114(15), pp.4023-4028.

Response: Thanks. The article has been cited where appropriate (ref 16).

REVIEWERS' COMMENTS

Reviewer #1 (Remarks to the Author):

I thank the authors very much for a thoroughly revised manuscript. Most of my comments have been addressed. I have three remaining comments: the first one is major, the other two are small and minor.

Coding of NPIs.

I understand the reasoning for summarizing NPIs into intensity levels to compare them in different contexts. I still think that the coding could be done differently, but I realize this may involve considerable effort during revision and may be too much to ask. Therefore, I start with presenting my suggested approach and then ask for two clarifications in case the authors would stick to their current approach of coding.

- Suggested approach.

First, I understand now that the proportion of the population affected cannot be used for weighting the NPI effects. However, a reasonable approximation would be to just use the geographical scope level like the way it is used in the computation of the normalized intensity score. That is, use as population weights $a_j / \max(a_j)$. This would allow factoring the geographical scope out of the NPI effects.

Second, to include different stringency levels of NPIs rather as binary indicator variables would be to decompose the NPIs effects into $\alpha = \alpha_s + \alpha_i + \alpha_c$ using a non-centered parametrization (see https://mc-stan.org/docs/2_18/stan-users-guide/reparameterization-section.html). Here α_s is the stringency effect (e.g. the Lockdown Level 1), α_i is the individual NPI effect (e.g. Lockdown), and α_c is the category effect (e.g. social distancing measures). The latter would be of particular interest to the authors considering their reporting of results. I realized that it may be difficult to fit such a model, especially in combination with varying slopes for variants. As a remedy, the authors can apply more informative priors for the variation in α_s and α_i , since they are anyway mainly interested in α_c .

The problem that I and (as it seems) the other reviewers have with the current coding is simply interpretability and comparability. With the above decomposition, the calculation of normalized intensity scores could be avoided. Moreover, the authors could directly explore variation in α_s and α_c , maybe revealing some additional interesting findings.

- Current approach

I can also understand if the authors would like to stick to their current approach. The aforementioned model is possibly not easy to fit and, to be fair, many other studies have resorted to intensity/stringency levels when analyzing NPIs. In case the authors stick to their current approach, I would just ask for two clarifications.

First, in the calculation of the intensity score (Supplementary Material A.2), is there a reason why x_j is calculated as $(I_j - 1) + a_j / \max(a_j)$ and not as $I_j / \max(I_j) + a_j / \max(a_j) - 1$? Since both I_j and a_j are reported on an ordinal scale, it seems more intuitive to normalize them the same way as well. Whether they should actually be normalized in the same way is a different story and relates to the issue of comparability with this approach in general.

Second, I understand that the average intensity level is used for each individual NPI. How is the effect then aggregated by category? I guess it is the sum of the $\alpha_j * \text{mean}(X_j)$ for each category, right? I apologize if I have overlooked it somewhere in the manuscript or supplementary material. Note that the authors could also use the median instead of the mean as the median would rather represent the intensity level that the population was exposed to most of the time.

Prior for NPI effects.

Note that prior by Flaman has been designed for 6 NPIs to generate a uniform prior for the total effect (see their Supplementary Material). As you note, the influence of the prior is probably minor. You could still run a quick check and compare the prior with the posterior distribution of the alphas, to see whether there is any indication of a strong prior vs data conflict (e.g. the prior being too restrictive).

Term "Lockdown".

Me (https://papers.ssrn.com/sol3/papers.cfm?abstract_id=4032477) and others (<https://www.cmaj.ca/content/195/15/E552.short>) believe that the term "lockdown" is, unfortunately, often misleading or misunderstood. Lockdowns can mean very different policies in different regions, so the people's perception of the policy varies considerably. I would suggest using the term "stay-at-home order" or "shelter-in-place order" in your case.

Reviewer #2 (Remarks to the Author):

Comments for authors, Nature Communications manuscript NCOMMS-XXXXXXX "Effects and challenges of public-health measures for zeroing out emerging contagions with varying transmissibility".

General comments:

Many thanks to the authors for a careful and responsive revision. The authors have addressed the conceptual points raised in my initial review, but some of the phrasing is awkward. I offer the following suggestions for clarity of presentation within the new text.

- Response to Specific Comment 2. Current text: "In China, MS and CT were defined differently as essential approaches for infection detection and tracing." Suggested clarification: "In China, both MS and CT were deemed essential approaches for infection detection and tracing but differ in definition and implementation as follows."

- Response to Specific Comment 3. Current text: "while social distancing was general requirements applied to all individuals". Suggested clarification: "while social distancing represents a general requirement applied to all individuals".

- Question: The authors' revised text reads: "Moreover, in places with considerable population flow, such as airports, contact tracing and mass screening might be more effective than social distancing and masking in reducing imported infections^{39,44}". References notwithstanding, I find it difficult to see how contact tracing and mass screening could be implemented in a location like an airport. Am I missing something?

- Response to Specific Comment on Supplement, line 299-307. In my original comment I was referring to the specific approach of "Approximate Bayesian Computation (ABC)" used to obtain posterior inference on model parameters for dynamic models (see Csilléry et al. 2010. "Approximate Bayesian Computation (ABC) in practice" Trends in Ecology and Evolution, or Marin et al. 2012. "Approximate Bayesian computational methods" Statistics and Computing). These use simulation of model outputs to define a likelihood so they are sometimes referred to as "likelihood-free" methods. I believe the authors were referring to "approximate Bayesian computing" in a general way rather than

in specific reference to ABC. If the reference is general, then I suggest modifying the new text "A Bayesian optimisation framework based on the approximate Bayes' computation was developed to optimize quantization hyperparameters in ISEIRV models" to "A Bayesian optimisation framework was developed to optimize quantization hyperparameters in ISEIRV models". If ABC methods were indeed used (building an approximate likelihood from repeated simulations from the ISEIRV model), I suggest providing one of the general references to ABC above.

Reviewer #3 (Remarks to the Author):

I appreciate the authors' efforts in addressing my questions. Congratulations on this interesting work.

Reviewer #1 (Remarks to the Author):

I thank the authors very much for a thoroughly revised manuscript. Most of my comments have been addressed. I have three remaining comments: the first one is major, the other two are small and minor.

Response: We are grateful for the reviewer's valuable suggestions and constructive comments, which significantly improved the quality and clarity of our manuscript.

Coding of NPIs.

I understand the reasoning for summarizing NPIs into intensity levels to compare them in different contexts. I still think that the coding could be done differently, but I realize this may involve considerable effort during revision and may be too much to ask. Therefore, I start with presenting my suggested approach and then ask for two clarifications in case the authors would stick to their current approach of coding.

- Suggested approach.

First, I understand now that the proportion of the population affected cannot be used for weighting the NPI effects. However, a reasonable approximation would be to just use the geographical scope level like the way it is used in the computation of the normalized intensity score. That is, use as population weights $a_j / \max(a_j)$. This would allow factoring the geographical scope out of the NPI effects.

Second, to include different stringency levels of NPIs rather as binary indicator variables would be to decompose the NPIs effects into $\alpha = \alpha_s + \alpha_i + \alpha_c$ using a non-centered parametrization (see https://mc-stan.org/docs/2_18/stan-users-guide/reparameterization-section.html). Here α_s is the stringency effect (e.g. the Lockdown Level 1), α_i is the individual NPI effect (e.g. Lockdown), and α_c is the category effect (e.g. social distancing measures). The latter would be of particular interest to the authors considering their reporting of results. I realized that it may be difficult to fit such a model, especially in combination with varying slopes for variants. As a remedy, the authors can apply more informative priors for the variation in α_s and α_i , since they are anyway mainly interested in α_c .

The problem that I and (as it seems) the other reviewers have with the current coding is simply interpretability and comparability. With the above decomposition, the calculation of normalized intensity scores could be avoided. Moreover, the authors could directly explore variation in α_s and α_c , maybe revealing some additional interesting findings.

Response: Thank you for providing this insightful suggestion. We understand the benefits of the proposed approaches. First, as the reviewer mentioned, our current approach to quantify the intensity (I_j) of NPIs has considered the relevant geographical scope (a_j). Since the effects are determined by the coefficients of NPIs in our model, the geographically adjusted NPI intensities allowed us to remove potential confounding effects of the geographical scope from the overall effects of NPIs that were implemented in different geographical ranges. However, we agree that it would be interesting to investigate the impact of geographical scope on the measurement and evaluation of NPI intensities and effects in future.

Second, the idea of splitting NPIs with the same stringency into different levels and independently observing their effects is really interesting, as it may enhance comparability and interpretability if we have sufficient data for this analysis. However, as the reviewer pointed out, it is quite challenging to build a model using many independent

variables for only 131 outbreaks. Categorizing the existing 10 NPIs into 31 intensity levels might introduce the complexity and raise concerns about overfitting. Therefore, for the current study, we have decided to retain the commonly used intensity/stringency levels approach and are genuinely open to explore this proposed method in future research.

- Current approach

I can also understand if the authors would like to stick to their current approach. The aforementioned model is possibly not easy to fit and, to be fair, many other studies have resorted to intensity/stringency levels when analyzing NPIs. In case the authors stick to their current approach, I would just ask for two clarifications.

First, in the calculation of the intensity score (Supplementary Material A.2), is there a reason why x_j is calculated as $(I_j - 1) + a_j / \max(a_j)$ and not as $I_j / \max(I_j) + a_j / \max(a_j) - 1$? Since both I_j and a_j are reported on an ordinal scale, it seems more intuitive to normalize them the same way as well. Whether they should actually be normalized in the same way is a different story and relates to the issue of comparability with this approach in general.

Response: The current approach ensures each normalized value corresponds to a specific and distinct intensity level-geographical scope combination, providing a more precise representation of the data. If we normalized I_j and a_j separately before adding them together, as you suggested, there is a possibility that the same value could represent multiple meanings, potentially leading to confusion and misinterpretation. For example, consider $\max(I_j) = 2$ and $\max(a_j) = 4$, following your proposed method, when the geographical scope level is 4 and the intensity level is 1, the normalized value would be $4/4 + 1/2 - 1 = 0.5$; and when the geographical scope level is 2 and the intensity level is 2, the normalized value would still be $2/4 + 2/2 - 1 = 0.5$. However, using our current method, for the former, the normalized value would be $[(1-1) + 4/4] = 1$; and for the latter, the normalized value would be $[(2-1) + 2/4] = 1.5$. This shows the potential ambiguity in the normalized values if we were to normalize I_j and a_j separately before adding them together. Thus, we believe our current approach provides a clearer and more specific representation of the intensity level-geographical scope combination. That is the intensity I_j determined the base value $I_j - 1$, and the geographical scope $\sim [0,1]$ give the final value $x_j \sim [I_j - 1, I_j]$ where I_j naturally represents the NPI was deployed in the whole area.

Second, I understand that the average intensity level is used for each individual NPI. How is the effect then aggregated by category? I guess it is the sum of the $\alpha_j * \text{mean}(X_j)$ for each category, right? I apologize if I have overlooked it somewhere in the manuscript or supplementary material. Note that the authors could also use the median instead of the mean as the median would rather represent the intensity level that the population was exposed to most of the time.

Response: Thank you for reminding an important point regarding the aggregation of NPI effects by category. We apologize for any confusion in our previous description. In our revised manuscript (Line 541-543), we have modified it as "***And the effect of NPIs can be computed by $1 - \exp(-\sum_i^n \alpha_{i,v} \bar{X}_{i,t,v})$, which $\bar{X}_{i,t,v}$ represents the median intensity of each individual NPI X_i for variant v . n represents the number of NPIs in each category. The median value of NPI intensity was calculated based on the normalized NPI.***" Additionally, we followed your suggestion to replace the mean by the median for aggregation, as it provides a more robust representation of the typical NPI intensity experienced by the population. Accordingly, we have updated the relevant results in our manuscript.

Prior for NPI effects.

Note that prior by Flaman has been designed for 6 NPIs to generate a uniform prior for the total effect (see their Supplementary Material). As you note, the influence of the prior is probably minor. You could still run a quick check and compare the prior with the posterior distribution of the alphas, to see whether there is any indication of a strong prior vs data conflict (e.g. the prior being too restrictive).

Response: Thank you. We have compared the prior with the posterior distribution of the alphas and shared the results in the form of a posterior distribution plot (see Supplementary figs 7-8), as you suggested.

Supplementary Fig 7. The posterior distribution of effect parameters of NPIs (a) stay-at-home order, (b) business premises closure, (c) public transportation closure, (d) gathering restriction, (e) workplace closure, (f) school closure, (g) facial masking, (h) mass screening, (i) medicine management, (j) facial masking, (k) contact tracing.

Supplementary Fig 8. The posterior distribution of the correlation parameter of the control factors (a) air temperature, (b) vaccine, (c) population density.

Term “Lockdown” .

Me (https://papers.ssrn.com/sol3/papers.cfm?abstract_id=4032477) and others (<https://www.cmaj.ca/content/195/15/E552.short>) believe that the term

“lockdown” is, unfortunately, often misleading or misunderstood. Lockdowns can mean very different policies in different regions, so the people’ s perception of the policy varies considerably. I would suggest using the term “stay-at-home order” or “shelter-in-place order” in your case.

Response: Agree, we renamed the term "Lockdown" as "Stay-at-home order" in our manuscript.

Reviewer #2 (Remarks to the Author):

Comments for authors, Nature Communications manuscript NCOMMS-XXXXXXX

“Effects and challenges of public-health measures for zeroing out emerging contagions with varying transmissibility” .

General comments:

Many thanks to the authors for a careful and responsive revision. The authors have addressed the conceptual points raised in my initial review, but some of the phrasing is awkward. I offer the following suggestions for clarity of presentation within the new text.

Response: Thank you for your positive feedback on our revision. The suggestions and comments you made have significantly enriched the content and presentation of our research.

- Response to Specific Comment 2. Current text: “In China, MS and CT were defined differently as essential approaches for infection detection and tracing.” Suggested clarification: “In China, both MS and CT were deemed essential approaches for infection detection and tracing but differ in definition and implementation as follows.”

Response: Agree. We have made the suggested change.

- Response to Specific Comment 3. Current text: “while social distancing was general requirements applied to all individuals” . Suggested clarification: “while social distancing represents a general requirement applied to all individuals” .

Response: We have modified this sentence as suggested.

- Question: The authors’ revised text reads: “Moreover, in places with considerable population flow, such as airports, contact tracing and mass screening might be more effective than social distancing and masking in reducing imported infections^{39,44}” . References notwithstanding, I find it difficult to see how contact tracing and mass screening could be implemented in a location like an airport. Am I missing something?

Response: We appreciate your comments on the feasibility of implementing contact tracing (CT) and mass screening (MS) at locations like airports. In the context of controlling the spread of COVID-19, the implementation of CT and MS has indeed been a common measure in China, particularly at densely populated areas like airports and railway stations. The Chinese government introduced a health QR code system during the pandemic, which utilizes big data to screen users’ travel history, PCR test results and identify their potential exposure to infected individuals, determining whether the user should be allowed to travel using public transportation or should be isolated as a close contact. Airports and railway stations are typically among the areas with the strictest enforcement of the health QR code system. Additionally, for inter-city and inter-province travel, airports, railway stations, and other similar transit points have set up nucleic acid testing stations, requiring all individuals to undergo COVID-19 testing within 48 or 24 hours before departure and, importantly, upon arrival. To avoid any misunderstanding, we have revised the manuscript to remove “such as airports” from the sentence.

- Response to Specific Comment on Supplement, line 299-307. In my original comment I was referring to the specific approach of “Approximate Bayesian Computation (ABC)” used to obtain posterior inference on model parameters for dynamic models (see Csilléry et al. 2010. “Approximate Bayesian Computation (ABC) in practice” Trends in

Ecology and Evolution, or Marin et al. 2012. “Approximate Bayesian computational methods” Statistics and Computing). These use simulation of model outputs to define a likelihood so they are sometimes referred to as “likelihood-free” methods. I believe the authors were referring to “approximate Bayesian computing” in a general way rather than in specific reference to ABC. If the reference is general, then I suggest modifying the new text “A Bayesian optimisation framework based on the approximate Bayes’ computation was developed to optimize quantization hyperparameters in ISEIRV models” to “A Bayesian optimisation framework was developed to optimize quantization hyperparameters in ISEIRV models” . If ABC methods were indeed used (building an approximate likelihood from repeated simulations from the ISEIRV model), I suggest providing one of the general references to ABC above.

Response: Thanks. The text has been modified as “A Bayesian optimisation framework was developed to optimize quantization hyperparameters in ISEIRV models”.

Reviewer #3 (Remarks to the Author):

I appreciate the authors' efforts in addressing my questions. Congratulations on this interesting work.

Response: Thank you.